



# Explaining changes in rainfall-runoff relationships during and after Australia's Millennium Drought: a community perspective

Keirnan Fowler[1], Murray Peel[1], Margarita Saft[1], Tim J. Peterson[2], Andrew Western[1], Lawrence Band[3,4], Cuan Petheram[5], Sandra Dharmadi[6], Kim Seong Tan[7], Lu Zhang[8], Patrick Lane[9], Anthony Kiem[10],
Lucy Marshall[11,12], Anne Griebel[13], Belinda E. Medlyn[13], Dongryeol Ryu[1], Giancarlo Bonotto[1], Conrad Wasko[1], Anna Ukkola[14], Clare Stephens[13], Andrew Frost[15], Hansini Gardiya Weligamage[1], Patricia Saco[10], Hongxing Zheng[8], Francis Chiew[8], Edoardo Daly[2], Glen Walker[16], R. Willem Vervoort[12], Justin Hughes[8], Luca Trotter[1], Brad Neal[17], Ian Cartwright[18], and Rory Nathan[1]

[1]Department of Infrastructure Engineering, The University of Melbourne, Parkville, Victoria, Australia
[2]Department of Civil Engineering, Monash University, Clayton, Victoria, Australia
[3]Department of Environmental Science, University of Virginia, Charlottesville, Virginia, USA
[4]Department of Engineering Systems and Environment, University of Virginia, Charlottesville, Virginia, USA
[5]CSIRO Land and Water, Sandy Bay, Tasmania, Australia
[6]Department of Environment, Land, Water and Planning, Melbourne, Victoria, Australia
[7]Melbourne Water, Docklands, Victoria, Australia
[8]CSIRO Land and Water, Black Mountain, Australian Capital Territory, Australia
[9]School of Ecosystem and Forest Sciences, University of Melbourne, Parkville, Victoria, Australia
[10]Centre for Water, Climate and Land (CWCL), College of Engineering, Science and Environment (CESE), University of Newcastle, Newcastle, New South Wales, Australia
[11]School of Civil and Environmental Engineering, University of New South Wales, Kensington, New South Wales, Australia
[12]ARC Training Centre Data Analytics for Resources and Environments, School of Life and Environmental Sciences, The University of Sydney, Camperdown, New South Wales, Australia
[13]Hawkesbury Institute for the Environment, Western Sydney University, Richmond, New South Wales, Australia
[14]Climate Change Research Centre, University of New South Wales, Kensington, New South Wales, Australia
[15]Bureau of Meteorology, Sydney, New South Wales, Australia
[16]Grounded In Water, Adelaide, South Australia, Australia
[17]Hydrology and Risk Consulting (HARC), Blackburn, Victoria, Australia
[18]School of Earth, Atmosphere and Environment, Monash University, Clayton, Victoria, Australia

*Correspondence to*: Keirnan Fowler (fowler.k@unimelb.edu.au)

**Abstract.** The Millennium Drought lasted more than a decade, and is notable for causing persistent shifts in the relationship between rainfall and runoff in many south-east Australian catchments. Research to date has successfully characterised where and when shifts occurred and explored relationships with potential drivers, but a convincing physical explanation for observed changes in catchment behaviour is still lacking. Originating from a large multi-disciplinary workshop, this paper presents a range of possible process explanations of flow response, and then evaluates these hypotheses against available
evidence. The hypotheses consider climatic forcing, vegetation, soil moisture dynamics, groundwater, and anthropogenic influence. The hypotheses are assessed against evidence both temporally (eg. why was the Millennium Drought different to previous droughts?) and spatially (eg. why did rainfall-runoff relationships shift in some catchments but not in others?). The results point to the unprecedented length of the drought as the primary climatic driver, paired with interrelated groundwater



processes, including: declines in groundwater storage, reduced recharge associated with vadose zone expansion, and reduced
connection between subsurface and surface water processes. Other causes include increased evaporative demand and
interception of runoff by small private dams. Finally, we discuss the need for long-term field monitoring, particularly
targeting internal catchment processes and subsurface dynamics. We recommend continued investment in understanding of
hydrological shifts, particularly given their relevance to water planning under climate variability and change.

# 1 Introduction

## 1.1 Importance of understanding hydrological response to multi-year drought

Change and variability are ubiquitous in environmental and human systems (eg. Milly et al., 2008; Wagener et al., 2010;
Montanari et al., 2013). In recent times, much effort has been spent understanding and modelling dynamic behaviour,
notably under the "change in hydrology and society" decade of the International Association of Hydrological Sciences for
2013-2022 (Montanari et al., 2013). While many short-term hydrological phenomena (e.g. flooding and seasonal transitions)
are relatively well-studied, long-term (multi-year) hydrological dynamics are only recently receiving increased attention.
This paper is concerned with hydrological response to multi-annual droughts, a topic recently explored on multiple
continents including North and South America, China and Australia (e.g. Avanzi et al., 2020; Fowler et al., 2020; Peterson et
al., 2021, Alvarez-Garreton et al., 2021).

With climate change, many parts of the world will experience long-term drying with more frequent and severe droughts (eg.
Cook et al., 2018). Therefore, it is particularly important to study historic multi-year droughts as the best available analogue
for such conditions. However, few informative examples of multi-year drought are available because of short streamflow
records which do not include historic droughts (eg. Verdon-Kidd and Kiem, 2009) and often are affected by lengthy gaps.
Lack of concurrent field measurements of different components of the catchment water balance (eg. shallow and deep
groundwater head, soil moisture, actual evapotranspiration) also limits our ability to explore and test alternative explanations
for observed behaviour. Likewise, these issues impede studies on drought recovery (eg. Peterson et al., 2021) and
distinguishing long-term changes from variability, particularly in highly variable climates (Morin, 2011).

For hydrologic modellers, it is difficult to be confident that modelling techniques are robust without past examples of
sustained changes in climate to test them on. There are many aspects of catchment behaviour that are not explicitly simulated
but which may prove salient to determine hydrologic response, such as transient ecosystem dynamics related to water stress
and regrowth, in addition to anthropogenic factors (eg. Van Loon et al., 2016; Stephens et al., 2021). Non-linear behaviour
and/or "tipping points" may be part of the response of a system to unseen climatic forcing (Rodriguez-Iturbe et al., 2009;
Peterson et al., 2009; 2021; Tauro, 2021). Acknowledging these issues, the ability for models to "extrapolate to changing
conditions" was recognised as one of 23 key challenges (unsolved problems) in hydrology by Blöschl et al. (2019). Even



models successfully tested against past hydroclimate data may still be unable to extrapolate predictions to the future under enhanced atmospheric $CO_2$ concentrations, higher temperatures, and potentially more frequent, severe and longer dry spells (Saft et al., 2016a; Fowler et al., 2018; Stephens et al., 2020). Nonetheless, testing against a range of historic conditions remains an important first step towards robust modelling.


Thus, examples of multi-year drought are valuable to hydrology, and Australia's Millennium Drought is one such example (van Dijk et al., 2013; Chiew et al., 2014). This paper focuses on the Millennium Drought as a case study, and it is hoped that lessons learned from this example are transferable and useful for understanding long-term hydrologic change in other places around the world. As explained in Section 1.2, this drought was notable for causing persistent shifts in the relationship between rainfall and runoff in many catchments (Saft et al., 2015) and other locations globally have since seen similar

behaviour, such as the United States (California; Avanzi et al., 2020), Chile (Garreaud et al., 2017; Alvarez-Garreton et al., 2021), and China (Tian et al., 2020). Future research should examine the underlying causes for all such cases, seeking lessons that are transferable in time and space to other regions that are yet to experience such shifts. With many regions of the world likely to see long-term drying, more frequent droughts and water scarcity (Schewe et al., 2014; Cook et al., 2018;

Kirono et al., 2020), it is critical to understand the causes of apparent hydrological non-stationarity and, ideally, incorporate this into operational models used by water planners.

## 1.2 The Millennium Drought

The Millennium Drought occurred in south-east Australia, affecting an area of more than 500,000 km². The drought commenced in the mid-to-late 1990s (1997 is taken as the start date for this paper) and is assumed to have ended with

widespread flooding in 2010-11 (starting in September 2010). It was primarily a rainfall deficit drought (Van Loon and Van Lanen, 2011), although it is noted that some average-rainfall-years occurred mid-way, in some places. Post 2011, climatic conditions have been similar to pre-drought averages in some places, and below average in others (Argent et al., 2017; Pepler et al. 2021).

The Millennium Drought was the longest drought on record in many affected areas and it caused significant social and economic impacts. The severe hydrological drought was estimated to have a return period exceeding 300 years (Potter et al., 2010; Freund et al., 2017). Water shortages induced low water use allocations for irrigators, water restrictions in urban areas, and significant investment in infrastructure such as desalination plants and pipelines (van Dijk et al., 2013). The economic costs of the drought have been estimated to be as much as 1.6% of Australia's GDP (Horridge et al., 2005). Concern for

environmental degradation led to significant water reforms, notably the Water Act (Connell and Grafton, 2011; Skinner and Langford, 2013), including a "cap and trade" -type system to restrict total water use in the Murray Darling Basin. This drought, and possibly projections of a drier future, accelerated major water reforms, such as the return of almost 20% of





irrigation water entitlements to the environment to be managed by new agencies called Environmental Water Holders (Skinner and Langford, 2013, Connell, 2011; Gawne et al., 2020).


Many catchments showed unexpected hydrological behaviours during the Millennium Drought, with streamflow being surprisingly low even when low rainfall was taken into account. This behaviour, referred to as a "shift" (Saft et al., 2015) or a "change in rainfall-runoff relationship" (Petheram et al. 2011), occurred in some catchments but not others (see Section 2), and many streams also became more intermittent as part of the transition and/or less saline (Kho et al., in review). Despite

the end of the meteorological drought in 2010, many catchments have not recovered, meaning that they have not shifted back to pre-drought behaviour and appear to be persisting in an alternate state (rather than simply slow to recover; Peterson et al., 2021). In shifted catchments that have not recovered, an average rainfall year produces less streamflow today than it did pre-drought.

The physical mechanisms underlying these shifts are not well understood. As discussed in Section 2, research to date has successfully characterised where and when shifts occurred (Petheram et al., 2011; Saft et al., 2015; Peterson et al., 2021) and explored statistical relationships with potential drivers (Saft et al., 2016b; Peterson et al., 2021). However, the scientific and management communities still lack a convincing and widely accepted physical explanation for observed changes in catchment behaviour, which is the subject of this paper.

**1.3 Context and aims of this article**

This article arose out of a virtual workshop in late 2020, where a multi-disciplinary group of scientists gathered to discuss the hydrological consequences of the Millennium Drought. Integrating perspectives from hydrogeology, plant ecophysiology, remote sensing, hydroclimatology, experimental hydrology and hydrological modelling, the workshop aimed to share and discuss hypotheses about catchment behaviour and associated flow response that could explain the Millennium

Drought streamflow observations, considering both the spatial and temporal patterns of hydrological shifts.

Synthesising and building upon the discussions in the workshop, this article aims to:
- Discuss relevant processes and how they could be responsible for observed shifts;
- Review available evidence and discuss strengths and limitations of that evidence;

- Consider whether a single holistic perceptual model of catchment response can be formulated by integrating the process explanations which are consistent with evidence; and
- Recommend future research and/or monitoring activities which may confirm, distinguish between and/or improve the process explanations.





Figure 1 shows the structure of the remainder of this article. Section 2 describes the hydrological shifts in greater detail, including a summary of key research to date. Section 3 considers a variety of processes relevant to streamflow generation, reviews the current state of knowledge in each case, and hypothesises process explanations based on this knowledge. Section 4 presents diagnostic evidence, tests the hypotheses against the evidence, and seeks to harmonise them into an integrated description of hydrologic response. Section 5 discusses future monitoring and research priorities required to further develop

and/or distinguish between the plausible process explanations.

Underlying this structure is the implicit assumption that all considered processes and hypotheses are potentially relevant to the scientific community, even in cases which turned out to be inconsistent with empirical data. An alternative would be to only present the final set of process explanations along with associated evidence, but we feel this would remove much of the

richness of the process of discovery and evaluation. Recent articles by Beven and Chappell (2021) and Wagener et al. (2021) affirm the value of documenting changes in qualitative understanding of catchment behaviour and the perceptual (mental) models we use to describe catchments (see also McGlynn et al., 2002). Sharing perceptual information (and opening it to wider evaluation) is particularly important when pushing the limits of hydrological knowledge (Wagener et al., 2021) and working in multidisciplinary contexts (eg. Staudinger et al., 2019). In any case, process explanations inconsistent with

empirical data may correctly explain the hydrology of drought events in other locations or at other times. Thus, we feel there is value in providing a full account of the hypotheses considered.

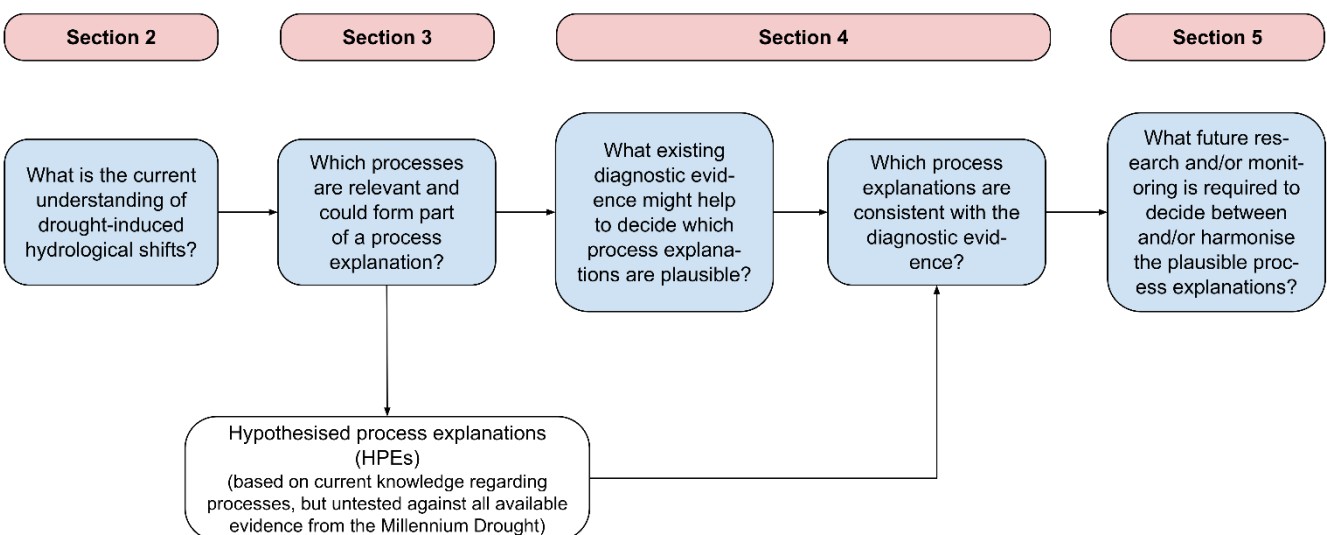

**Figure 1: Key questions for each section in this article**



## 2. Current understanding of drought-induced hydrological shifts

The purpose of this section is to describe the current understanding of the Millennium-Drought-induced hydrological shifts. To provide a concise summary, we focus on two papers which most directly examine the issue (namely Saft et al., 2015 and Peterson et al., 2021) while acknowledging that other papers exist exploring related themes (eg. Petheram et al., 2011; Westra et al., 2014; Saft et al., 2016a, b; Fowler et al., 2016; 2020; Ajami et al., 2017; Deb et al., 2019) including elsewhere in Australia (Petrone et al., 2010; Hughes et al., 2012) and globally (see introduction). Both papers examine the rainfall-runoff relationships using the annual timestep, which reduces the impact of seasonal storage differences and smooths out within-year extremes. For this section only, we focus on the annual timestep as a way of introducing the shifts in rainfall-runoff relationships, including the temporal and spatial scale of the changes. However, as seasonal and event dynamics may be a crucial element of process explanations, they are considered at length in Section 3.

The essence of the phenomenon is that the hydrological response during the drought was out of proportion to the change in rainfall over the same period—in other words, the streamflow during the Millennium Drought was lower than expected even after the lower rainfall is taken into account (Figure 2a-b). For example, Saft et al. (2015) demonstrated statistically significant shifts in the annual rainfall-runoff relationship when considered separately over the Millennium Drought period and the pre-drought period. The implication is that a given annual rainfall led to a smaller annual streamflow during the drought than before. Approximately half of the studied catchments shifted (Figure 2c), especially drier and flatter catchments with low forest cover. The shifts were often associated with an increase in the number of zero-flow days.

The shifted behaviour has continued post-drought. Using more recent data, Peterson et al. (2021) showed that more than half of the shifted catchments had not yet recovered by 2017 (Figure 2d), despite a return to wetter conditions particularly in 2010, 2011 and 2016. In non-recovered catchments, years with average rainfall produce less streamflow today than they did pre-1997. Peterson et al. (2021) found that the catchments appear to be persisting within a low runoff state and that non-recovery is not explained by land cover change or the rating curve or gauge method. While this study was over a smaller extent (state of Victoria, Fig. 2c-d), the non-recovery was found to be best explained by increased ET per unit of precipitation, rather than increased interception, groundwater storage or recharge.

The shifted catchments had experienced drought previously during the period of streamflow record, as reflected by the low pre-1997 rainfall years in Figure 2b. However, there was something different about the Millennium Drought that caused streamflow response to diminish relative to previous droughts. Having experienced prior years of low rainfall, this set up an expectation of how much streamflow could be expected in a dry year of given severity (measured by rainfall), and this expectation was subsequently broken during (and after) the Millennium Drought.



**Figure 2. Hydrological behaviour prior to and during the Millennium Drought (a-c) and post-drought recovery (d), based on prior studies. (a) annual timeseries of precipitation (P) and streamflow (Q) for four selected catchments which exhibited hydrological shifts; (b) scatter plots corresponding to (a); (c) location map of shifted catchments, from Saft et al. (2015); (d) drought recovery with time, mapped across the state of Victoria for selected time-slices from Peterson et al. (2021). Elapsed time is measured from the end of the meteorological drought.**



Finally, there are some important points of difference with other studies examining hydrological drought. Some studies use
the word "drought" to refer to periods of seasonal low flow during dry seasons and not extending into the following wet
season. In contrast, this article is concerned with multi-year drought, so the drought impacts analysed are year-round. Indeed,
any hypothesis seeking to explain the multi-annual changes reported above must inherently also explain behaviour during the
wettest times of year, since these are responsible for most of the streamflow.

## 3. Possible process explanations

This section presents process explanations that were hypothesised by workshop attendees, while Section 4 evaluates these
explanations against evidence. Thus, presentation of evidence is mostly held over to Section 4, with the exception of
evidence from prior literature, some of which is presented in plots in this section. Numerous categories of processes are
considered, each in a separate subsection, and explanations involving more than one category are marked accordingly and
grouped with the most important category for the mechanism in focus. For clarity, Hypothesised Process Explanations are
labelled with the acronym "HPE" followed by an identification number and a one sentence summary, starting with
*"Streamflow was lower than expected because…"*. Table 1 below summarises all the HPEs.

### 3.1 Changes in meteorological conditions

The problem framing in Section 2 already accounts for changes in precipitation, but considered annual values only, not
within-year changes, such as sequencing and seasonality, nor changes in variables other than precipitation. These changes
are the focus below.

### 3.1.1. Changes in rainfall seasonality

Earlier studies such as Potter and Chiew (2011) suggested seasonality of rainfall as one causative factor for lower
Millennium Drought streamflows. As seen in Figure 3a, precipitation reduced across the seasons, and to different degrees in
different areas, but reductions in autumn (March-May) were the most severe in many areas. There were also reductions in
winter rainfall (see also Pepler et al., 2021), although the percentage reductions were not as great as those in autumn.

In this study area, autumn is the time when many catchments "wet up" ahead of the main flow generating period in winter
(June-August) and early spring. In terms of hydrological processes, the "two water worlds" hypothesis (Brooks et al., 2009;
McDonnell et al., 2014) confirms the importance of autumn rainfall. This concept differentiates between tightly bound soil
water (used by vegetation) and water destined for streams and lakes. Isotope studies in Mediterranean climates have shown
that these sometimes do not mix, and that the tightly bound soil water, once seasonally depleted, gets priority in the seasonal





**Table 1. Summary table of hypothesised process explanations (HPEs).**
*Note: HPEs marked with \* also assume declines in recharge, eg. due to lower precipitation and/or HPEs 6-7, 9, or 13-15*

| | HPE | Short Name | Streamflow was lower than expected because… |
|---|---|---|---|
| **Meteorological conditions** | 1 | Rainfall seasonality | lower autumn rainfall led to delayed "wetting up" and muted flow response during winter (when runoff coefficients are usually highest). |
| | 2 | Rainfall dry spells | more time spent in spells without rain led to higher initial losses of event rainfall and less conversion to streamflow. |
| | 3 | Fewer large events | fewer rainfall events exceeded thresholds to produce runoff. |
| | 4 | Evaporative demand | evaporative demand increased at a time of year when catchments are relatively less water limited and relatively more energy limited. |
| | 5 | Long drought | expectations were conditioned by shorter droughts which didn't trigger long-term or long-memory process responses. |
| **Vegetation dynamics** | 6 | Greening | fertilisation by higher $CO_2$ concentrations caused higher vegetation water use |
| | 7 | Coupling & radiation / temp. | higher radiation and/or air temperature increased transpiration, particularly in grassland catchments. |
| | 8 | Delayed desiccation | higher $CO_2$ concentrations allowed grasslands to delay seasonal desiccation and transpire water for longer. |
| | 9 | Species replacement | vegetation mortality led to species replacement with plant types that are more opportunistic and thus use more water. |
| | 10 | Tree regulation of water use (stomatal control) | *assuming some other mechanism for lower streamflow*, stomatal control by trees (responding to high VPD) made extra water available for streamflow, counteracting this mechanism in forests. |
| | 11 | Tree regulation of water use (other) | *assuming some other mechanism for lower streamflow*, water use controls such as xylem cavitation or reduced leaf area made extra water available for streamflow, counteracting this mechanism in forests |
| | 12 | Wildfire | *assuming some other mechanism for lower streamflow*, the occurrence of forest wildfires made extra water available for streamflow, counteracting this mechanism in forests. |
| | 13 | Salinity recovery | plant health (and thus water use) recovered, and new trees were added, in areas formerly affected by waterlogging and/or salinity. |
| | 14 | Grassland litter | accumulated decomposing litter in grasslands intercepted more precipitation, causing higher AET. |
| | 15 | Biophysical adaptation | biophysical adaptation to the disturbance of the Millennium Drought caused an increase in the fraction of precipitation going to transpiration. |
| **Groundwater & subsurface** | 16 | Sustained hillslope drainage* | sustained groundwater drainage from hillslopes (but not across catchment boundaries) contributed to declines in the water table and reduced GW-SW interaction. |
| | 17 | Sustained regional drainage* | sustained drainage of groundwater across catchment boundaries contributed to declines in the water table and reduced GW-SW interaction. |
| | 18 | Gaining to losing | stream reaches lost more water (or gained less water) due to water table decline. |
| | 19 | Vadose zone available capacity | vadose zone interception increased due to its thickening and drying. |
| | 20 | Soil property changes | drought-induced changes in soil properties such as higher porosity and/or cracking caused higher infiltration and evaporation. |
| **Anthropogenic** | 21 | Agricultural practices | better management of agricultural land, including soil conservation techniques, caused differences in soil hydraulic properties. |
| | 22 | Plantation forestry | increased plantation forestry led to increased water use. |
| | 23 | Farm dams | farm dam interception increased due to trends in dam development (ie. more dams than prior droughts) |
| | 24 | Groundwater extraction | groundwater extraction caused declines in groundwater levels and reduced GW-SW interaction. |



**Figure 3. Change in selected climatic statistics during the Millennium Drought\* compared to the long term average\*\* for each season. Each dot corresponds to a single catchment; values plotted are areal averages sourced from Fowler et al. (2021). \*taken as April 1997 to August 2010, inclusive. \*\*taken as calendar years 1950 through 2017, inclusive.**

re-wetting process (Brooks et al., 2009). This means that water can move to the stream only after the soil pores have been replenished. During the Millennium Drought, lower autumn rainfalls may have extended the soil-pore-filling period later into the year, diminishing the period of flow generation, a concept supported by Wasko et al. (2020). The Hypothesised



Process Explanation (HPE) is thus summarised as: ***HPE01: Streamflow was lower than expected because lower autumn rainfall led to delayed "wetting up" and muted flow response during winter (when runoff coefficients are usually highest).***

### 3.1.2. Increased time spent in spells of low rainfall

As shown in Figure 3b, during the Millennium Drought there was an increased proportion of time spent in spells with little or no rainfall in many locations, although it is noted by Verdon-Kidd and Kiem (2009) that this increase was not as high as during two other protracted droughts pre-1950. Nonetheless, we retain it as a hypothesis for now: ***HPE02: Streamflow was lower than expected because more time spent in spells without rain led to higher initial losses of event rainfall and less conversion to streamflow.***

### 3.1.3. Decreases in frequency of high-rainfall events

Generally, a small number of large rainfall events each year contribute disproportionately to annual flow volumes, and there is recent evidence for continent-wide reduction in large rainfall events (Wasko et al., 2019; Dey et al., 2020). Heavier rainfall events are more conducive to preferential flow, but smaller or less intense events may be more readily absorbed by the soil and thus available to plants (eg. Elliot et al., 2015). A reduction in the frequency (Figure 3c) of high rainfall events (ie. those above thresholds that produce runoff* - Saffarpour et al., 2016) could produce a downward shift in streamflow generation even if annual reductions in precipitation are relatively moderate. Furthermore, we might expect that flatter catchments require higher rainfall thresholds to produce runoff than steep catchments; it is thus possible that we might see differences in response even within a homogenous rainfall region. Supporting this hypothesis, Verdon-Kidd and Kiem (2009) reported that the Millennium Drought had less intense rainfall, on average, than two other protracted droughts pre-1950. ***HPE03: Streamflow was lower than expected because fewer rainfall events exceeded thresholds to produce runoff.*** *Note: ideally Figure 3c would plot the data against catchment-specific thresholds of runoff production, but this varies temporally and is difficult to quantify. Thus, a simple threshold is used instead, as marked in Figure 3c.

### 3.1.4. Increases in evaporative demand

Stephens et al. (2018) reported recent increases in pan evaporation within the area affected by the Millennium Drought (despite earlier declines), while Potter and Chiew (2011) cited higher evaporative demand as a potential driver of hydrological behaviour. Although south-east Australia is typically water limited rather than energy limited, anomolies in evaporative demand may still be important seasonally and in wetter parts of the landscape. Potential evapotranspiration (PET) increases when water is available (eg. winter and spring is the wettest time in the southern part of the affected area) are more likely to impact the water balance. Figure 3d shows that evaporative demand—here quantified as Morton's (1986) "Wet Environment" evaporation—increased significantly at precisely this time of year, and more so in the drier south and south west of the study area (in central and western Victoria, cf. Fig. 2). Conspicuously, this is also an area where almost all



catchments exhibit hydrological shifts (Fig. 2c-d). The question of which environmental variable drives this increase is held
over to Section 4 (see also HPE07). In summary, ***HPE04: Streamflow was lower than expected because evaporative
demand increased at a time of year when catchments are relatively less water limited and relatively more energy limited.***

### 3.1.5. Length of drought

The Millennium Drought lasted 10, 13 or 16 years depending on location and choice of definition. By comparison, many
earlier droughts, such as 1965-68 and 1982-83, were shorter. It is possible that the unexpectedly severe streamflow
reductions during the Millennium Drought were caused by the activation of long-term or long-memory processes that were
absent from these earlier droughts. The pre-1950 period included relatively longer droughts such as the Federation Drought
(1895–1903) and the World War 2 Drought (1939–1945), and while comparison of meteorological conditions among these
droughts is insightful (eg. Verdon-Kidd and Kiem, 2009), few streamflow gauges were active during these periods (see also
Section 4.1) which makes intercomparison of flow difficult. ***HPE05: Streamflow was lower than expected because***
***expectations were conditioned by shorter droughts which didn't trigger long-term or long-memory process responses.***
This broad, overarching HPE is incomplete without further detail as to which processes are in view (see later HPEs).

### 3.1.6. Comments on climatic drivers

The climate drivers presented above—namely PET, rainfall seasonality, dry spells and intensity—all relate to the
Millennium Drought period, not post-drought. As many catchments did not recover post-drought, a test of the veracity of
these HPEs is whether the stated climatic driver continued post-drought or went back to pre-drought conditions. For
example, did the changes in seasonality cease with the drought, or continue post-drought? We leave such questions to
Section 4 as part of the evaluation of the HPEs.

## 3.2. Vegetation dynamics

In south-east Australia, high evaporative demand relative to rainfall means that evapotranspiration dominates the water
balance, and rainfall-runoff ratios are low (typically <10%). Therefore, vegetation behaviour has the potential to significantly
impact streamflow, as a small percentage change in plant water availability or AET may translate into a large percentage
change in streamflow. Recent findings suggesting an increase in AET per unit of precipitation (Peterson et al. 2021; see also
Sawada and Koike, 2016) underscore the need for investigation.

### 3.2.1. Greening induced by increased atmospheric $CO_2$

Atmospheric $CO_2$ concentrations have been increasing over the instrumental period (IPCC, 2021), with their effect on
vegetation water use and impact on catchment or regional water resources being the subject of considerable research (eg.
Peel et al., 2009, Cheng et al. 2018). Higher atmospheric $CO_2$ concentrations generally increase plant water use efficiency





(e.g. Ainsworth et al. 2005; Gedney et al., 2006; Liu et al., 2020), which may manifest as a reduction in transpiration (e.g.

Warren et al. 2011). However, the increase in water use efficiency can also manifest as increased growth, and in some cases this can lead to increased transpiration (e.g. Piao et al., 2007; Uddling et al. 2008). Observational studies from eastern Australia supporting the latter view include Ukkola et al. (2016), Ajami et al. (2017) and Transcoso et al. (2017). This suggests a potential cause for the reduced streamflow, but it is noted that responses may depend on landcover, eg., forests and grasslands (Morgan et al., 2004; 2011) (see next subsection) and may also vary under drought conditions (Yang et al.,

2016). It should also be noted that, because $CO_2$ fertilisation typically leads to a reduction in transpiration per unit leaf area, increased growth can often occur without any increase in transpiration (De Kauwe et al 2021; Rifai et al. 2021). ***HPE06: Streamflow was lower than expected because fertilisation by higher $CO_2$ concentrations caused higher vegetation water use*** (relative to previous droughts).

### 3.2.2. Atmospheric coupling, vapour pressure deficit & evaporative demand

This HPE spans vegetation behaviour, climatic forcing (HPE04) and atmospheric physics. Atmospheric coupling refers to the degree of mixing of air between the canopy and the atmosphere above, which is related to the turbulent flow over and within the canopy (McNaughton and Jarvis, 1991). Small-stature canopies (eg. grasslands) are more decoupled due to lower turbulence associated with homogeneous canopies (eg. De Kauwe et al., 2017). This lower turbulence in grasslands allows a near-surface boundary layer to build up, which decouples the vapour pressure deficit (VPD) experienced by plants from the

atmospheric VPD. Thus, transpiration in grasslands is relatively less responsive to atmospheric VPD and relatively more responsive to other factors such as solar radiation and temperature. ***HPE07: Streamflow was lower than expected because higher radiation and/or air temperature increased transpiration, particularly in grassland catchments.*** Discussion of whether increases in radiation actually occurred (and with a spatial distribution supporting this hypothesis) is held over to Section 4. Note, in contrast to grasslands, transpiration from forests in south-east Australia can be very sensitive to

atmospheric VPD, and this is discussed further in the next subsection.

### 3.2.3. Moisture stress, delayed desiccation and vegetation replacement

Under low moisture, plants may become water stressed, and leaves (or more rarely, the whole plant) may die. In grasslands with pronounced dry seasons, loss of leaves due to desiccation may occur seasonally leading to reduced water use, but the root system typically remains alive despite loss of foliage. Higher $CO_2$ allows more efficient use of available water in

grasslands (eg. Hovenden et al., 2019) and may delay seasonal desiccation in C4 grasses under warmer conditions (Morgan et al., 2004; see also Morgan et al., 2011). Paradoxically, grasses that retain foliage for longer may access rainfall from mid-to-late summer storms, thereby intercepting more water than grass where leaves are already dead. This is a paradox because the higher water use efficiency, which enables the longevity of the leaves, could actually result in greater annual water use. ***HPE08: Streamflow was lower than expected because higher $CO_2$ concentrations allowed grasslands to delay seasonal***

***desiccation and transpire water for longer (relative to previous droughts).***


If vegetation dies entirely (eg. Allen and Breshears, 1998; Fensham et al., 2005, Semple et al., 2010) this could increase water use if the vegetation is replaced by species that are more opportunistic in their water use. ***HPE09: Streamflow was lower than expected because vegetation mortality led to species replacement with plant types that are more opportunistic***
***and thus use more water.***

As noted, transpiration from forests in south-east Australia can be very sensitive to atmospheric VPD (eg. Gharun et al., 2013), with plants exercising strong stomatal control to limit moisture loss during periods of high VPD and low soil moisture (eg. Renchon et al., 2018; Grossiord et al., 2020). Thus, if some other mechanism was acting to cause hydrological shifts
across grasslands and forests alike, then stomatal control in forests may have counteracted this mechanism (in which case forest catchments would appear less shifted than grasslands). ***HPE10: Assuming some other mechanism for lower streamflow, stomatal control by trees (responding to high VPD) made extra water available for streamflow, counteracting this mechanism in forests.*** Discussion of whether increases in VPD actually occurred (and with a spatial distribution supporting this hypothesis) is held over to Section 4.

Trees demonstrate additional water saving strategies (eg. McDowell et al., 2008) not related to stomatal control, with eucalypts being able to reduce leaf area (Nolan et al., 2021) or undergo xylem cavitation (sometimes temporarily) thus impairing water use (Blackman et al., 2019; Nolan et al., 2021). These are listed separately to the previous HPE because they may persist for longer than rapid stomatal dynamics and be less dependent on VPD. ***HPE11: Assuming some other***
***mechanism for lower streamflow, water use controls such as xylem cavitation or reduced leaf area made extra water available for streamflow, counteracting this mechanism in forests .***

### 3.2.4. Wildfire

Wildfire recovery is known to affect water use, with impacts subject to competing factors. Tree mortality and morbidity reduce water use, whereas re-growth increases it, and this is species-specific (Heath et al., 2014). Due to this combination of
factors, yield impacts reported by Australian studies vary widely and include significant increases (Lane et al. 2006; Guo et al., 2021), decreases in the short-medium term (Nolan et al., 2015) and longer term (Kuczera, 1987), and negligible change in catchment yield (Feikema et al., 2013; Heath et al., 2014). Many eucalypts have post-fire resprouting mechanisms (eg. epicormic buds, lignotubers) which can be activated provided the fire does not kill the tree (eg. Nolan et al., 2014). Thus, fire severity is an important factor, as is species mix (Nolan et al., 2015). A further consideration is changes to soil hydraulic
properties and water repellency following fire (eg. Nyman et al., 2010). South-east Australia was subject to multiple fire events during the Millennium Drought. For example, the 2003 bushfires were the first widespread fire events since the early 1980s, and subsequent major fires occurred in the summers of 2006-7 and 2008-9, over differing areas in each case. In general, the areas affected by wildfire were those that did not shift. The hypothesis is as follows: assuming a dip in





vegetation water use after the wildfire (eg. Lane et al., 2006, Guo et al., 2021), possibly coupled with runoff-inducing
changes in soil properties (eg. Nyman et al., 2010), the wildfire caused an increase in streamflow that masked the underlying
shift in behaviour. In other words, if not for wildfire, more catchments would have appeared to have shifted. ***HPE12:
Assuming some other mechanism for lower streamflow, the occurrence of forest wildfires made extra water available for
streamflow, counteracting this mechanism in forests***

### 3.2.5. Vegetation recovery from waterlogging and salinity

This HPE spans vegetation and groundwater processes. Prior to the Millennium Drought, multi-decadal wet conditions
combined with historic deforestation led to high water tables and waterlogging in many areas, with naturally-occurring soil
salt brought to the surface (eg. Cartwright et al., 2004). The waterlogged and saline topsoil reduced the health, and thus
water use, of vegetation (Lambers, 2003). The Millennium Drought lowered water tables and removed these stressors, aided
by tree-planting as part of salinity management efforts (Schofield, 1992). ***HPE13: Streamflow was lower than expected
because plant health (and thus water use) recovered, and new trees were added, in areas formerly affected by
waterlogging and/or salinity.***

### 3.2.6. Increased interception due to decomposing material

The drier conditions during the Millennium Drought, together with potential greening, may have led to a greater
accumulated dead plant matter in grasslands. Litter in grasslands has been shown to increase interception of precipitation
(eg. Naeth et al., 1991). ***HPE14: Streamflow was lower than expected because accumulated decomposing litter in
grasslands intercepted more precipitation, causing higher AET.***

### 3.2.7. Biophysical adaptation to climatic disturbance

Peterson et al. (2021) suggest vegetation behaviour may explain the apparent appearance of hydrological shifts (which they
term "multiple stable states" in rainfall-runoff behaviour), and the persistence of the low runoff state post-drought. They say:
"Evidence suggests that the vegetation responded to the drought by increasing the fraction of precipitation going to
transpiration… like other natural systems with multiple stable states, this persistence may be caused by a biophysical
adaptation to disturbances… that results in a positive feedback" (ibid. p4-5). The exact nature of this biophysical adaptation
is yet to be determined, and could be related to one or more of the preceding HPEs. In any case, given AET is the largest
element (after rainfall) of the water balance in south-east Australia, it seems reasonable to explain the hydrological shifts by
hypothesising a direct change to the dynamics of AET by some cause. If it is accepted that the vegetation is the primary
cause, this explains both the streamflow shifts (since less water is available for runoff) and the water table decline described
in the next subsection (since a higher fraction of drought rainfall is transpired, which dries out the soil, reducing recharge).
***HPE15: Streamflow was lower than expected because biophysical adaptation to the disturbance of the Millennium
Drought caused an increase in the fraction of precipitation going to transpiration.***




### 3.3. Groundwater and subsurface processes

As discussed above (HPE05) it has been hypothesised that the long duration of the Millennium Drought activated long-term (long-memory) processes. Given groundwater is a relatively large store of water that can exhibit multi-year trends and long memory, it seems reasonable to assume streamflow declines were related to groundwater—a hypothesis strengthened by
observed regional-scale decline in the water table in south-east Australia throughout the Millennium Drought (Figure 4; LeBlanc et al., 2011; van Dijk et al., 2013; Fowler et al., 2020). For this hypothesis to work, it is first necessary to establish that groundwater influences streamflow generation, as briefly summarised here. In headwater catchments, groundwater plays a role in streamflow generation through a variety of mechanisms. Shallower and deeper groundwater processes can act simultaneously and may have different response times to disturbance. For example, streamflow may arise from relatively
shallow interactions such as perched saturation along hydraulic gradient fronts (eg. at the transition from A to B horizons)— behaviour which is relatively transient, with short (sub-annual; days-weeks) response times (eg. Hughes et al., 2007). Interactions with unconfined aquifers include direct groundwater discharge including via springs, diffuse discharge areas, or directly into streams (eg. O'Grady et al, 2010; Cartwright and Gilfedder, 2015, Zhu et al., 2021). Also, diffuse discharge areas and areas with shallow water table create wet areas primed for event runoff (eg. Dunne and Black, 1970).


These varied processes—referred to hereafter as groundwater-surface water (GW-SW) interaction—mean that unconfined aquifers can interact and contribute to short timescale (event-to-seasonal) hydrological processes in headwater catchments. However, unconfined aquifers also exhibit slower dynamics such as multi-year trends. These dynamics depend on conditions: for high rainfall headwater catchments with steep slopes, lag times will be less than a few years, while for
medium rainfall lower slope catchments the time scale can be decades, depending on geology (Cook, 2003). The observed drought-induced multi-year water table decline (Figure 4a) likely caused discharge areas to dry up and seasonal wet areas to contract, thus providing mechanisms for flow reductions in headwater streams. In addition, groundwater-induced contractions in stream network length and extent increased subsurface flow path lengths to the nearest stream, potentially increasing the probability of interception by transpiration (Jensen et al., 2017; 2018). Together, the above points establish the
connection between groundwater and streamflow generation in headwater streams, leaving HPEs 16, 17, 19 and 20 focus on the cause of the groundwater decline. Note:

- Some of the vegetation-themed HPEs presented above (specifically HPEs 6-9, 13-15) could also reduce recharge and are thus also possible causes or contributors to groundwater decline. Even in the absence of these HPEs, it is reasonable to assume reduced recharge during the Millennium Drought since it is highly correlated with rainfall (eg.
Ferdowsian et al., 2001; Peterson and Western, 2014); and

- The effect of groundwater on streamflow is not limited to headwater streams (Section 3.3.3) and can involve extensive groundwater systems (Section 3.3.2).





## (a) Groundwater and GRACE records

## (b) Groundwater water balance conceptualisation

**Figure 4. (a) Groundwater hydrographs for nineteen bores and GRACE data in the state of Victoria, after Fowler et al. (2020); (b) water balance conceptualisation for unconfined aquifers in headwater catchments.**





### 3.3.1. Sustained local groundwater drainage from hillslopes

Multi-year water table declines in response to the Millennium Drought imply that the depleting fluxes of the groundwater water balance (see Figure 4b) are less sensitive to the multi-year dry conditions (ie. more sustained) than fluxes adding to it.

This HPE explains the depletion of groundwater by sustained underground drainage from hillslopes to local low points in the landscape (eg. base of slope, valley bottoms). These low points often contain deposits of alluvium (river deposits) and/or colluvium (unconsolidated sediments deposited at the base of hillslopes by gravity). Separate treatment of hillslope and alluvial aquifers has precedent in local groundwater-surface water studies such as the salinity-focussed rainfall-runoff modelling by Stenson et al. (2011). According to this HPE, the alluvial/colluvial deposits receive the drained water, and then

evaporation and transpiration within these deposits keep the water table below the surface and ensure much of this drainage does not contribute to streamflow. This drainage is hypothesised to occur continuously throughout the pre-drought, drought and post-drought periods. Pre-drought, the drainage was balanced by higher recharge, meaning that the water table in hillslopes was closer to the surface. This earlier shallow groundwater facilitated runoff processes in hillslopes that were subsequently diminished when the water table dropped, contributing to the hydrological shifts. During the drought, recharge

declined while, according to this HPE, the drainage was assumed to be more sustained, resulting in lower groundwater heads particularly in hillslopes. The plausibility of this HPE assumes lateral aquifer transmissivity does not decline so much with lower head as to shut off the lateral drainage from the hillslopes. Thus, this HPE depends on the presence of (i) geological strata that support the sustained drainage (eg. deeply fractured rock—Rempe and Dietrich, 2018), and (ii) colluvial/alluvial deposits (flat valley bottoms, not erosional/v-shaped). ***HPE16: Streamflow was lower than expected because sustained***

***groundwater drainage from hillslopes (but not across catchment boundaries) contributed to declines in the water table and reduced GW-SW interaction.*** The words "but not across catchment boundaries" are added to distinguish between this HPE and HPE17 (below).

### 3.3.2. Groundwater export across catchment boundaries

Whereas the previous HPE assumed drainage within a catchment, we can also hypothesise drainage crossing boundaries of

surface water catchments as part of a regional flow system (also known as "mountain block recharge" but in this case associated with hills rather than mountains) flowing inland, or alternatively discharging to the ocean. As before, the drainage is hypothesised to occur continuously throughout the pre-drought, drought and post-drought periods, with the drought bringing changes in the balance with other elements of the water balance in Figure 4b. ***HPE17: Streamflow was lower than expected because sustained drainage of groundwater across catchment boundaries contributed to declines in the water***

***table and reduced GW-SW interaction.*** We note that:

- In the context of headwater catchments, the distinction between local and regional drainage is important because:
  (i) regional discharge cannot support within-catchment "wetting up", which would otherwise amplify event runoff;



and (ii) at the point of discharge, groundwater discharge may be an important control on AET, so the distinction affects the appearance of a headwater catchment in remotely sensed data. Both (i) and (ii) affect how the catchment

460         might be represented in simulation models.

- A further distinction is that some systems, particularly those that are more extensive, tend to have longer response times to disturbance. Thus, combinations are possible whereby the drainage (or other groundwater dynamic) may have a mixture of dominant timescales—say, a 1-5 year timescale associated with a local system, mixed with a 10-30 year timescale associated with a regional system. Likewise, drought recovery may be a question of the relative

465         influence of shorter and longer timescale groundwater systems.

### 3.3.3. Gaining and losing streams

Even if rainfall-runoff relationships remained unchanged (ie. no shift) within lower-order catchments, flow within higher order stream reaches can be intercepted as it travels downstream, via seepage into the surrounding substrate. This is called a "losing" stream reach, in contrast to a "gaining" reach in which the stream receives water instead of losing it. The

losing/gaining dynamic is driven by the stream level relative to the groundwater level in the surrounding substrate. This dynamic may be important in cases where the river flows over extensive alluvial deposits, since (i) these deposits can store significant amounts of water; and (ii) they commonly have relatively high hydraulic conductivity, which may assist in dispersing the water horizontally over the extent of the deposit, creating opportunities for transpiration or evaporation (eg. Salama et al., 1993). Reaches can change from gaining to losing state and vice versa (often associated with seasonal cycles)

and they experience a net gain or loss while in a given state—a key point of difference with bank storage, which functions as a delaying process (instead of a source/sink term) as water flows into and then back out of stream banks as gradients switch back and forth episodically. Losing/gaining dynamics could have contributed to the observed shifts at stream gauges. The regional groundwater declines could have affected the spatiotemporal extent of losing reaches (ie. greater total length losing; greater time spent losing rather than gaining), amplified water loss rates from streams, and increased the number of zero flow

days. ***HPE18: Streamflow was lower than expected because stream reaches lost more water (or gained less water) due to water table decline.*** Note, while losses are sensitive to the water table, the reverse is also true, with losses from the stream causing the water table to increase, or decline less slowly than it would otherwise (subject to other fluxes such as evaporation and transpiration from the alluvium). The strength of this feedback/coupling increases for smaller alluvial stores.

### 3.3.4. Increase in vadose zone available capacity increase, via expansion and drying

Regardless of the cause of the water table decline, the result was a thicker vadose (unsaturated) zone that was likely drier than in pre-drought conditions. A thicker and drier vadose zone has enhanced potential to intercept rainfall, thus adding a further driver of recharge reductions. In soils, hydraulic conductivity declines steeply and non-linearly with wetness, so recharge may be much slower to get to the water-table. Overall, the result is an amplification of the hydrological effect of





other causes of vadose zone thickening and/or drying. ***HPE19: Streamflow was lower than expected because vadose zone***
***interception increased due to its thickening and drying.***

### 3.3.5. Changes in soil properties and/or structure with time

Soils may change properties with time, and some drought-related changes may decrease streamflow. While many soil
properties and temporal changes are driven by humans (discussed below: HPE21), there are other mechanisms that are not
explicitly human-driven, although they may interact with land management. For example, soil cracking may lead to
increased infiltration, more evaporation and less runoff (eg. Arnold et al., 2005). There is also evidence that soil porosity
may increase as climate gets drier, on relatively short timescales of years to decades (Hirmas et al., 2018; Caplan et al.,
2019). ***HPE20: Streamflow was lower than expected because drought-induced changes in soil properties such as higher***
***porosity and/or cracking led to higher infiltration and evaporation.*** On the other hand, soil water repellency can build up
during dry periods (eg. Filipovic, 2018), which would tend to increase streamflow, which is counter to observations.


### 3.4. Anthropogenic factors

### 3.4.1. Changes in agricultural practices

The last fifty years have seen changes in land management practices, and the hydrological impacts of these changes may be
important. Soil loss around Australia in the 1960s and 70s prompted a nation-wide evaluation of land degradation and soil
conservation strategies in the 1970s, in addition to a National Soil Conservation Program, established in 1983 (Hannam,
2001). As a result, soil conservation measures such as no-till farming (eg. Cornish et al., 2020) were more prevalent by the
1990s than previous decades, which may have caused hydrological differences (see eg. Huang and Zhang, 2004) between the
Millennium Drought and previous droughts. In general, poor management practices lead to degraded soils that may amplify
event runoff, with partial soil recovery (towards, eg. higher soil carbon) possibly causing trends towards higher infiltration.
Techniques such as contour banks may also retain water in the landscape. Lastly, soil properties were impacted by trends in
livestock stocking rates driven by both market forces and government policy. ***HPE21: Streamflow was lower than expected***
***because better management of agricultural land, including soil conservation techniques, caused differences in soil***
***hydraulic properties*** (compared to previous droughts).

### 3.4.2. Land-use change and plantation forestry

Land-use change is commonly reported to alter hydrology (eg. Zhang et al., 2001; Brown et al., 2005; Zhang and Schilling,
2006; Farley et al., 2005). In south-east Australia, a key example is new plantation forestry, with higher water demand of
plantation trees impacting groundwater (Benyon and Doody, 2004; Lane et al., 2005; Webb et al., 2012; Brown et al., 2015;
Dresel et al., 2018) and presumably streamflow. Plantation data (Legg et al., 2021) shows a peak in nation-wide plantations





in the late 1990s, with approximately 100,000 ha per year added for 1998-2000, much of it within the area impacted by the
Millennium Drought (Legg et al., 2021, Table 1). ***HPE22: Streamflow was lower than expected because increased plantation forestry led to increased water use.*** The 1970-90s also saw planting for salinity mitigation—see HPE13 above.

### 3.4.3. Farm dams

Farm dams, also known as farm ponds, are small private dams located on hillsides, drainage lines or small streams. Individual dams typically harvest only small volumes, but collectively the impact can be large (Habets et al., 2018; Morden
et al., 2021). Volumetrically, water harvesting by farm dams is greatest during moderate and high flows, but if expressed in relative terms, impacts are greatest during the dry season and during dry years (Morden et al., 2021). Farm dam development has increased with time (with a higher rate of increase before 2000—Malerba et al., 2021), so there were more dams during the Millennium Drought than previous droughts. **HPE23: Streamflow was lower than expected because farm dam interception increased due to trends in farm dam development (ie. more dams than prior droughts).**

### 3.4.4. Extractions

Regionally, the bulk of agricultural water use occurs in broad, flat plains using water captured in large reservoirs. This activity occurs downstream of the headwater catchments in focus here. However, minor modes of extraction are relevant in the headwater catchments, including extractions from groundwater and diversions from streams by individual landholders. For the set of catchments used by Peterson et al. (2021) and shown in Figure 2, individual diversions from streams are
uncommon because they were considered as a criterion in catchment selection, and where they do exist they are strictly controlled. Extractions from groundwater can be relatively larger than those from streams within some of these catchments, and are of interest given the strong groundwater trends reported above. Most extraction bores are for stock and domestic use. However, licences for irrigation, while rarer, typically correspond to larger volumes, and thus may be significant. **HPE24: Streamflow was lower than expected because groundwater extraction caused declines in groundwater levels and**
**reduced GW-SW interaction.**

### 3.5. Interaction between process explanations

The set of HPEs are mostly complementary, meaning there is no inherent reason precluding co-occurrence of multiple HPEs simultaneously in the same catchment. Some HPEs exhibit dependence, such as HPE15 ("gaining to losing"), which assumes the regional groundwater decline a priori and thus is dependent on a separate explanation for the decline via one of the other
HPEs. Likewise, HPE19 amplifies effects of other mechanisms. HPEs 10 and 11 (relating to tree regulation of water use) and 12 (relating to wildfire) are in a unique category: they posit mechanisms that may have masked underlying shifts, rather than mechanisms of the shifts themselves. Such HPEs may be important to explain spatial patterns in hydrological shifts. Overall, assuming the evidence tests in the next section do not eliminate too many HPEs, the complementary nature of the HPEs seems to indicate that we should expect a combination of process explanations, rather than a single "smoking gun".





**4. Evidence-based critique of hypotheses**

As per Figure 1, the purpose of this section is to decide which hypothesised process explanations are plausible or more likely, based on consideration of diagnostic evidence. This section is subdivided as follows: Section 4.1 explains the logic of evaluation; Section 4.2 presents the diagnostic evidence, assesses the implications for HPEs, and compares findings with previous literature; Section 4.3 attempts to harmonise the plausible HPEs to provide an integrated picture of catchment

behaviour; and Section 4.4 discusses the limitations of the deductive method used.

**4.1. Logic of evaluation**

There are four ways in which HPEs are assessed for consistency with the evidence, denoted with symbols ($\triangledown$; $\triangle$; $\diamond$; $\square$) as indicated below. While in principle it would be preferable that any HPE declared plausible would be assessed in all of these ways, in practice it is often impossible to do so due to lack of data, particularly going back in time.

1. **Temporal consistency ($\triangledown$):** the focus is on comparing droughts across time, focussing on the Millennium Drought and prior droughts. Since prior droughts set our expectations for what flow to expect in a dry year, there must be something different about the Millennium Drought that caused the streamflow to be lower than expected (in those catchments that shifted). Those droughts that occurred before streamflow gauging commenced have not contributed to our expectations of what flow is 'normal' in a drought year. Thus, the prior droughts under consideration are

those that occurred after a subjectively selected date (1st January 1950) that nominally represents the start of streamflow gauging. The year 1950 is chosen because less than 20% of streamflow gauges were operational at that time (Figure 6); hence results from studies such as Saft et al. (2015) and Peterson et al. (2021) were dominated by post-1950 data.

   2. **Spatial consistency ($\triangle$):** the focus is on the spatial distribution of the hypothesised driving factor(s). To support the

hypothesis, the spatial distribution of the driving factor should be consistent with the spatial distribution of shifted versus unshifted catchments.

   3. **Water balance considerations ($\diamond$):** the focus is on whether the hypothesised driving factor is sufficient to reduce flow volumes by the observed amount. This criterion is less critical because the observed flow reductions may have been the combined result of multiple HPEs (in which case no single HPE would cause sufficient reductions on its

own).

   4. **Drought recovery ($\square$):** the focus is on whether the hypothesis can explain the lack of post-drought recovery in most of the catchments that shifted (Figure 2d). As per Peterson et al. (2021): "Across all 161 watersheds, we found that 8 years into the drought, 51% of watersheds switched into a low (or very low) runoff state. When the drought ended in 2010, predominantly only the eastern watersheds shifted back to a normal-runoff state. Notably, 7 years

after the drought, 37% (n = 55) of watersheds remained within a low-runoff state".





Two different spatial scales are considered in the following section. When assessing changes in climatic variables, we consider the entire area affected by the Millennium Drought, consistent with earlier consideration of climatic variables (Figure 3). However, for all other assessments, we concentrate on the southern state of Victoria ($2.3 \times 105$ km²; Figure 2d), for three reasons:

- **Existing analyses of hydrological shifts:** at the scale of Victoria, there are two recent studies (Peterson et al., 2021; Saft et al., under review) that examine hydrological shifts using every suitable streamflow gauge and using data up to 2017. In contrast, at the wider scale, the only available study (Saft et al., 2015) examined a dataset with lower density of spatial coverage and using data only up to 2008, being sufficient to assess impacts during the drought but not afterwards.

- **Data availability:** certain datasets, for example farm dams (Figure 8d), are available in a homogenous dataset across Victoria, but such data were not available in New South Wales and Queensland at the time of analysis.

- **Spatial patterns of impact:** as shown in Figure 2, Victoria has the clearest spatial groupings of shifted versus non-shifted catchments, with the latter dominating in the east and the former dominating in the west. These groupings arise despite considerable similarities between east and west: for example, the streamflow regime is winter-dominant in both cases. Starting the process of explaining the shift in Victoria, where we at least see some spatial organisation of shifted/non-shifted catchments, makes sense. Later work needs to extend this to the other states.

### 4.2. Evaluation of diagnostic evidence

The evaluation of individual HPEs is undertaken in Table 2, based on the variety of diagnostic evidence presented in Figures 5 - 8. Of the twenty-four HPEs, three are considered plausible, ten are considered inconsistent with evidence, and eleven are in a category in-between, denoting HPEs that may be important but with their widespread plausibility or applicability questioned by factors such as (i) limited spatial extent, (ii) limited water balance impact, and (iii) limited ability to explain post-drought non-recovery in rainfall-runoff relationship.

### 4.3. Harmonising plausible process explanations

This subsection seeks to combine the plausible process explanations into an integrated picture of landscape-scale hydrological response. The following description, depicted visually in Figures 9 and 10, incorporates every process explanation marked with green or yellow in Table 2. Given the focus of Section 4.2 on evidence from the southern state of Victoria, the following applies to the same area, although many elements may apply to other affected areas.

Concerning climatic conditions, two things distinguish the Millennium Drought from prior droughts post-1950:

1. **Longer drought:** The drought lasted more than 10 years (Figure 6; HPE05), allowing long-memory aspects of catchment behaviour to become important, as discussed below;

2. **Evaporative demand increases:** Over a significant area in central-west Victoria, the evaporative demand



**Table 2. Assessment of Hypothesised Process Explanations (HPEs), evaluating consistency with evidence.**

| | HPE | Short Name | | Supported by evidence? (green = yes (y); red = no (n); amber = maybe/partly/in some cases (m)) Assessment types: ▽ = temporal; △ = spatial; ◇ = water balance; □ = drought recovery. |
|---|---|---|---|---|
| **Meteorological conditions** | 1 | Rainfall seasonality | n | The Millennium Drought did not have greater autumn reductions in rainfall than prior droughts (▽, Fig. 5a), the spatial patterns of rainfall reductions were not skewed towards shifted catchments (△, Fig. 3a), and lower autumn rainfall has not continued post-drought (□, Fig. C1); this HPE on its own cannot explain the hydrologic shifts. This is consistent with Peterson et al. (2021) who analysed seasonal precipitation to assess if changes to seasonal flow explains non-recovery, and concluded that it did not explain the hydrological shifts post drought |
| | 2 | Rainfall dry spells | n | The Millennium Drought did not have longer dry spells than prior droughts (▽, Fig. 5b), the spatial patterns of spell behaviour were not skewed towards shifted catchments (△, Fig. 3b), and longer spells have not continued post-drought (□, Fig. C2); this HPE on its own cannot explain the hydrologic shifts. |
| | 3 | Fewer large events | n | The Millennium Drought did not have more severe reductions than prior droughts in the number of large rainfall events (▽, Fig. 5c), the spatial patterns were not skewed towards shifted catchments (△, Fig. 3b), and significant reductions have not continued post-drought (□, Fig. C3); this HPE on its own cannot explain hydrologic shifts. |
| | 4 | Evaporative demand | m | The Millennium Drought had higher PET during spring (Sept.-Nov.) than prior droughts (▽, Fig. 5d) over a significant area (the area south of line α-α' in Fig. 5d) and particularly in shifted catchments in central-west Victoria (△, yellow dots in Fig. 5d-i). On the other hand, this occurred over a limited spatial area, and continued post-drought but with a different spatial distribution (more towards the south-east of Australia; □; Fig. SI1). Working with annual (not seasonal as here) data, Saft et al (2015) reported a difference in PET anomaly between shifted and unshifted catchments, with a p-value of 0.08, which while not significant at the α=0.05 is a low probability of occurring by chance. |
| | 5 | Long drought | y | Prior droughts during the 1950s-1980s were typically 1-3 years (▽; Fig. 6). Longer droughts occurred earlier (eg. 1890s & 1940s), but rare flow monitoring means that those droughts have not contributed to expectations for what is 'normal' streamflow during drought. |
| **Vegetation dynamics** | 6 | Greening | m | Greening has the potential for significant effects; eg. Ukkola et al. (2016) estimate $CO_2$ changes over 1982-2010 may have caused average AET increases of 43 mm/yr (sub-humid) and 14 mm/yr (semi-arid), which would be a significant proportion of streamflow (which is typically less than 100 mm/yr in shifted catchments; ◇). Furthermore, Ukkola et al. (2016) analysed temporal trends in Normalised Difference Vegetation Index (NDVI) across forested catchments, and the results could be interpreted to show a spatial pattern across Victoria, with higher temporal trends among shifted catchments (see their Figure S4). Sawada et al. (2016) found positive temporal trends in NDVI and negative trends in Vegetation Optical Depth in areas subject to hydrological shifts. In both cases, interpretation is difficult because the spatial coverage is relatively poor (few catchments for the former study; blocky and gappy in the latter) and also many of the trends were not statistically significant. In general, few existing studies directly address the problem at hand: Ukkola et al. (2016) examine only forested catchments (whereas shifted catchments are more commonly grasslands - Fig. 7c); Sawada et al. (2016) have patchy coverage; Ajami et al. (2017) examine only a handful of catchments; and Trancoso et al. (2017) examine baseflow only. Furthermore, in $CO_2$ experiments, it is uncommon to see increased transpiration: LAI increases are accompanied by a decrease in stomatal conductance, such that the effects compensate for one another (Yang et al., 2018; Uddin et al., 2019; De Kauwe et al., 2021; Fitzgerald et al., 2021; Rifai et al., 2022). More research is required, with focus on grasslands and historic $CO_2$. |
| | 7 | Coupling & radiation / temperature | m | Analysis of the drivers of PET (Fig. B1-B2) reveals that radiation was responsible for the spatial patterns (△) of higher PET (relative to previous droughts, ▽) during spring (HPE4) in southern catchments. The higher radiation did not continue post-drought (□, Fig. C5), although $t_{min}$ was higher (Fig. C6). |
| | 8 | Delayed desiccation | n | The southern part of the study area is dominated by C3 grasses (△) although C4 are common further north (Munroe et al., 2021; Xie et al., 2022). In any case, it is doubtful whether modest historical $CO_2$ increases could have caused large changes in the water balance via this mechanism (◇). |
| | 9 | Species | n | There were minimal reports of climate-induced mortality, let alone replacement, in headwater catchments during |





| | HPE | Short Name | | Supported by evidence? (green = yes (y); red = no (n); amber = maybe/partly/in some cases (m)) Assessment types: ▽ = temporal; △ = spatial; ◇ = water balance; □ = drought recovery. |
|---|---|---|---|---|
| | | replacement | (red) | the Millennium Drought, although more recent droughts may be different (De Kauwe et al., 2020). Shifted catchments are typically agricultural (△), and in such areas the climate-driven species change of forested catchments does not apply (see also HPE21). |
| | 10 | Tree regulation of water use (stomatal control) | m | Analysis of the drivers of PET (Fig. B1-B2) reveals that Millennium Drought VPD, while higher than the long-term average, was typically equal to or lower than VPD in prior droughts (▽). However, post-drought VPD has been significantly higher across south-east Australia (□, Fig. C7; Stephens et al., 2018; Denson et al., 2021; Abram et al., 2021), so this may influence recovery. |
| | 11 | Tree regulation of water use (other) | m | The ability of trees to regulate water use, including through reduced leaf area and temporary xylem cavitation, is a key element of water partitioning in forested catchments. Shifted catchments tended to be dominated by grasslands (Fig. 7c), which lack such ability . Thus, such regulation may help to explain differences in partitioning between grasslands and forests (△), but it is uncertain the extent to which these effects can persist for multiple years after the end of a drought, which is relevant to recovery (□). |
| | 12 | Wildfire | n | As shown in Fig. 8a, it was mainly non-shifted catchments affected by wildfires during, and immediately prior to, the Millennium Drought (△). However, many non-shifted catchments were minimally affected (21 of 32 catchments shown in Fig. D2 were less than 20% affected) with only a small minority (2 of 32) more than 40% affected by the fires (Fig. D2). These statistics lump all fire severities, so lightly burned areas are included. Furthermore, if wildfires were masking underlying shifts, we would expect affected catchments to appear shifted up until the time of burning, and then shift abruptly back to a "normal" flow state. This was not observed in the time-varying Hidden Markov modelling of Peterson et al. (2021). |
| | 13 | Salinity recovery | m | Salinised land is more common among shifted catchments (△; Fig. 8b), and effects would continue post-drought given the water table has remained low (□; Peterson et al. 2021 Fig. S17-S18), so this HPE is plausible. The fraction of land affected is typically less than 2% among shifted catchments (Fig. 8b), which is likely insufficient on its own to cause the observed shifts (◇). |
| | 14 | Grassland litter | n | Although grass litter has been reported as a significant control in colder climates (Naeth et al., 1991), decomposition rates are accelerated by the warmer temperatures of south-east Australia (eg. Couˆteaux et al., 1995). Further, in areas with Mediterranean climates, seasonal grass die-off was near-complete even prior to the Millennium Drought, and thus the drought did not necessarily accelerate litter accumulation. |
| | 15 | Biophysical adaptation | m | Vegetation-induced AET shifts neatly explain streamflow decreases and water table declines. However, it is unclear what specific ecophysiological mechanism is responsible. Shifted catchments tend to be pasture dominated (△), so an ecophysiological mechanism would need to either (i) focus on adaptation of grasses; (ii) focus on trees and explain how the hydrological changes arise from relatively few trees in the landscape; or (iii) a mixture of these two. |
| Groundwater & subsurface | 16 | Sustained hillslope drainage | y | Duff et al. (1982) state that fractured rock groundwater systems underlay extensive areas co-located with shifting catchments (△). Spatial products such as the Multi-Resolution Valley Bottom Flatness (MRVBF, Gallant et al., 2003) indicate a higher level of deposition in valley bottoms in the areas containing shifted catchments (△). Thus, spatially (△), both components of this HPE are supported. Temporally, prior droughts (post 1950) were too short for sustained groundwater decline, thus providing a differentiating factor for the Millennium Drought (▽). Post drought, the lower GW-SW interaction would be expected to last until the water table recovers (□), explaining post-drought non-recovery. Further, spatial differences in recovery vs non-recovery could be explained by GW system response times. |
| Groundwater & subsurface | 17 | Sustained regional drainage | m | This HPE is supported temporally (▽) and in terms of post-drought recovery (□) for the same reasons as HPE16. Spatially (△), it is plausible in many, but not all, shifted catchments, since mapping of groundwater systems (Fig. 7d) shows approximately one third of shifting catchments (and all non-shifted catchments) are within the "upland valley" system category, which is unlikely to be co-located with regional groundwater systems (DSE, 2012). For the other two thirds of shifted catchments, this HPE is plausible. |





| | H P E | Short Name | | Supported by evidence? (green = yes (y); red = no (n); amber = maybe/partly/in some cases (m)) Assessment types: ▽ = temporal; △ = spatial; ◇ = water balance; □ = drought recovery. |
|---|---|---|---|---|
| | 18 | Gaining to losing | m | This HPE is supported temporally (▽) for the same reasons as HPE16. This HPE requires a relatively large alluvial store to receive the stream seepage. Smaller alluvial stores will be filled more easily, leading to more rapid equilibrium and no further losses. Large alluvial stores are more likely in lowland areas away from the "upland valley" GW system category (△, DSE, 2012; Fig. 7d), which describes some but not all shifted catchments. Post-drought (□), renewed supply of runoff from upstream would be expected to refill the alluvial store, leading to recovery of the water table in the alluvial store. However, this has rarely been observed, with a falling, or low but constant, head after the drought being more common (Peterson et al. 2021), although it is noted existing studies do not categorise bores according to whether they are within the alluvial store. |
| | 19 | Vadose zone available capacity | y | This HPE is supported temporally (▽) and in terms of post-drought recovery (□) for the same reasons as HPE16. Groundwater decline was widespread across the areas of Victoria containing shifted catchments (△, Fig. 4). It is considered likely that groundwater head decline initiated by other factors would result in a vadose zone better able to intercept water and further reduce recharge, thus amplifying the effect. |
| | 20 | Soil property changes | m | In areas with Mediterranean climates, it is expected that soil cracking would have occurred seasonally during previous droughts and, in many cases, during wet years also. Thus, it is not a differentiating aspect for the Millennium Drought (▽). In the literature, observed at-site porosity changes on short timescales appear to be limited to very different landscapes than encountered in the study area (eg. upland heath in Wales, UK in Robinson et al., 2018); thus, based on current evidence, this mechanism does not seem likely, but is worth attention in future research (cf. Hirmas et al., 2018). |
| Anthropogenic | 21 | Agricultural practices | m | Agriculture dominates in shifted catchments but is rarer in non-shifted catchments (△, Fig. 7c). The timing of onset of trends towards better land management fits: ie. the period of greatest change was the two decades leading up to the Millennium Drought (▽; see Section 3.4.1). Further, assuming permanent behaviour change, effects could continue to the post-drought period (□). We currently lack the information to estimate hydrological impacts (◇), including how they vary across space. |
| Anthropogenic | 22 | Plantation forestry | n | Although the area of additional plantations is large (eg. Victoria's forestry increased 20%, or 65,000 ha from 1999 to 2004; Legg et al., 2021 Table 1) a mapping of the current spatial distribution of plantations (Fig. D3) indicates a lack of overlap with the study catchments used by Peterson et al. (2021), with most in the far south-west corner of Victoria. Peterson et al. (2021) also looked at land cover change comparing time-series land cover data with their runoff states. Land cover change did occur (their Fig S8) but wasn't related to runoff state (their Fig S9 and S10). Consistent with conclusions in Peterson et al. (2021), plantations are not considered a likely cause of changes in such study catchments (△), although they are likely important to local hydrological changes in some places. |
| Anthropogenic | 23 | Farm dams | m | Farm dams are common in shifted catchments and less common in non-shifted catchments (△; Fig. 8d). Construction rates were high in the years leading up to the Millennium Drought (▽; Malerba et al., 2021) and dams added at that time would be expected to continue impacting flow up until the present time, consistent with lack of post-drought recovery (□). The remaining question is the magnitude of impacts (◇). Annual flow impacts of farm dams vary but are typically on parity with their volume (SKM et al., 2010). Thus, annual flow impacts (Fig. 8d) can be relativised against flow by converting total estimated volume per unit area to an equivalent depth, and then expressing this depth as a % of local mean annual flow. Although this metric exceeds 50% within the drier parts of some shifted catchments, when assessed at the catchment scale only 29 of 119 shifted catchments exceed 10%; and only 12 exceed 20% (◇). , These figures relate to total farm dams, not the increments between droughts, so they are an upper envelope. Given these low figures, it is unlikely farm dams are a dominant cause of hydrological shifts, but are clearly contributing. |
| Anthropogenic | 24 | Groundwater extraction | n | Although extractions are more common among shifted catchments than non-shifted (△), there are few catchments where extraction is large compared to streamflow (◇, Figure 8c). While locally relevant to a handful (<5) of catchments, this HPE is not supported broadly. |

615

**Figure 5.** Changes compared to the long term average (1950-2017) for (i) the Millennium Drought; and (ii) prior droughts (selected from the period 1950-1996 inclusive), allowing HPEs to be assessed by temporal consistency (∇). Variables and definitions are the same as Figure 3. "Prior droughts" refers to a set of years chosen separately for each catchment from the period 1950-1996. Years are selected if their rainfall was less than the average over the Millennium Drought (1997-2009). If this selects more than one quarter of years in the period 1950-1996, the selection is limited to the driest quartile (12 years). This selection process chooses a different number of years for each catchment, as shown in Figure D1. Line α-α' is mentioned in Table 2.



**Figure 6. (a, c) Visualisation of the longer duration of the Millennium Drought and World War 2 Drought compared to other droughts in the 1950s-1980s by showing anomalies in annual precipitation; (b) flow gauging coverage through time. Precipitation and flow data sourced from Fowler et al. (2020).**

625





**Figure 7. Maps of the state of Victoria showing various attributes used to assess HPEs by their spatial consistency (△). Each map is shown with 156 catchments categorised by Saft et al. (under review). Note, a given location in space can be marked with more than one Saft et al. category due to nestedness (ie. if it is upstream of two gauges in different categories). Data sources are listed in the supplement.**





**Figure 8. Similar to Figure 7 but mapping different variables. Relativisation by annual flow for (c) and (d) uses areal flow estimates from Weligamage et al. (2021). Other data sources are listed in the supplement.**





**Figure 9. Visual synthesis of selected process explanations explaining why the Millennium Drought had less streamflow than expected, relative to prior droughts (post-1950; these droughts were too short for significant groundwater decline). The situation depicted describes a typical catchment in central or western Victoria.**

increased, particularly during Spring (Figure 5c; HPE04). This increase was higher than increases seen in prior droughts (post-1950), and was driven by higher radiation. The affected area matches very closely with regional patterns of shifting versus not shifting. Given that the area is majority pasture, the nature of the increase contributed to its hydrologic impact, since grasslands are relatively less sensitive to atmospheric VPD and relatively more sensitive to radiation (see HPE07). Lastly, the time of year (spring, when soils are wettest) likely maximised the impact on the water balance.

Other aspects of climatic conditions, such as autumn rainfall reductions, less large rainfall events and longer dry spells (HPEs 1, 2 and 3 respectively), likely contributed to the lower streamflow but are considered less explanatory of the shifting







**Figure 10. Chain of causality from cause (top) to effect (bottom) for plausible hypothesised process explanations. Pure causes are at the top and pure effects are at the bottom, while those in between are a mixture (ie. effects of some things and causes of others).**

650    behaviour. This is because these aspects were no worse, and in some cases considerably less severe, than other drought years post 1950. However, the effect of these aspects may have been amplified due to (i) the unusual duration of the drought; and (ii) the fact they were happening in concert with (and contributing to) an interannual drop in water storage (negative water balance). Thus, while none of HPE1-3 are marked as explanatory in their own right, they may still be important pieces of the water balance puzzle.

655



In the combined perceptual model presented in Figures 9 and 10, the lower water table is a key driver of lower streamflow. This is because groundwater is integral to flow generation in headwater catchments, via numerous mechanisms (Figure 10). On hillslopes, groundwater declines reduce episodic saturation of high conductivity near-surface layers and/or transient perched water tables. In broad colluvial and alluvial valley bottoms, water table declines reduce flow in springs and diffuse-area discharge. Shallow groundwater creates wet areas in the landscape primed for event runoff, but such areas contract under water table decline. Thus, it is hypothesised that the unexpectedly low streamflow was in large part due to the declining water table, which was itself caused by:

1. **Recharge reductions.** Recharge reduced partly due to rainfall declines, and also due to increased evapotranspiration resulting from higher evaporative demand (HPE04) and possibly other changes associated with vegetation (HPEs 7, 13, 15).

2. **Sustained removal of groundwater via multiple fluxes.** The fluxes removing groundwater were less sensitive to the climatic conditions, and quite sustained (Figure 4b). These fluxes included:

   a. Local (within-catchment) drainage (HPE16) from hillslopes to base of slope and valley bottoms (Figure 9b) via deep fractures in the bedrock. Having drained via the subsurface to colluvial or alluvial valley floors where the water table is shallower (Figure 9b), this water can then be removed by transpiration or direct soil evaporation, which also helps to explain the increase in AET per unit of precipitation emphasised by Peterson et al. (2021). According to this HPE, most of the groundwater decline takes place in the hillslopes, which are capable of slow but significant drainage due to, eg., fractured rock aquifers, which are widespread in parts of Victoria with shifted catchments. Likewise, colluvial and alluvial deposits (refer HPE16 in Table 2) are more common in shifted catchments, whereas non-shifting catchments tend to be erosional with "v-shaped" valleys (Gallant et al., 2003; CSIRO, 2016).

   b. Drainage into regional groundwater systems (HPE17; Figure 9a, commonly known as "mountain recharge") which may export the groundwater outside of the boundary of the surface water catchment. Spatial applicability of this export mechanism is lower because approximately one third of shifted catchments are located away from regional systems (see HPE17 in Table 2 and Figure 7d).

   It is important to recognise that drainage in (a) and (b) above were not activated by the drought, but rather are hypothesised to be continuous, and the drought acted to change the balance that previously existed between these combined fluxes and recharge.

3. **Amplification effect due to vadose zone interception:** as groundwater drained, the increasingly thicker and drier vadose zone was better able to intercept infiltrated water, which amplified the recharge reductions (HPE16).





Furthermore, streamflow generated at the micro-catchment scale was likely susceptible to losses within downstream reaches. Lowland reaches which previously seasonally gained groundwater may now lose groundwater year-round (HPE18), with potential for subsequent removal from the alluvial store via AET. Water stored in river banks, which is gradually released on
timescales up to a few years and thus capable of sustaining low flows over an isolated dry year, became more permanently depleted.

Lastly, anthropogenic factors also likely impacted on hydrological shifts. The increase in farm dam development with time meant that the Millennium Drought saw greater farm dam interception than previous droughts (HPE23); however, the
magnitude of this impact is uncertain. Furthermore, the period prior to the drought saw improvements in land management that may have resulted in less degraded soils (eg. greater soil carbon) better able to retain water, with less rainfall partitioned towards event runoff or recharge.

All of the above mechanisms, except those related to climate, display some degree of persistence (see asterisked items in
Figure 10), and this may explain the persistent non-recovery that has been reported (Peterson et al., 2021) in some areas. In particular, groundwater-related mechanisms would persist until the recovery of the water table to pre-drought levels. This is likely to be spatially uneven as different groundwater systems have different lag-times and dynamics. Thus, a possible research avenue might be to compare perceived lag times (based on hydrogeological knowledge) with observed recovery of streamflow response.

Most mountainous study catchments in Victoria did not exhibit hydrologic shifts and this may be explained by the absence of the above mechanisms in those catchments, as follows:
- Such catchments tend to have erosional 'v' shaped valley; flat valleys arising from colluvial or alluvial deposits are relatively rarer (CSIRO, 2016), as are aquifers large enough to receive significant seepage from a losing reach;
- Regional groundwater flow systems are typically less common in such catchments (Fig. 7d);
- Salinity impacts were relatively low (or absent) in such catchments (Figure 8b);
- Farm dam development is very low (or absent) in such catchments (Figure 8d).
- Spring PET increases did not affect such catchments, nor catchments further north in New South Wales and Queensland (Figure 4c).

In addition, mountainous catchments are mostly forested, (Figure 7a and 7c) and some trees have mechanisms for reducing water use in times of water stress, such as temporary xylem cavitation and reducing leaf area (see Section 3.2.3). Tree water use is very sensitive to VPD and may steeply decline in times of high VPD. Such mechanisms, which alter partitioning within the water balance, may have contributed to the lack of shifting behaviour in forested catchments. Post-drought,
uniformly high VPD across south-east Australia (Fig SI1) may likewise be contributing to a lack of hydrological shifts in





forested catchments, as trees regulate water use via stomatal control. Lastly, while some research has examined the impact of rising $CO_2$ on streamflow (Ukkola et al., 2016), this prior work focussed exclusively on forested catchments whereas the hydrological shifts were associated with grasslands. It is possible that greening contributed to reduced streamflow in such catchments (HPE06), but historic greening effects in south-east Australian grasslands is currently a knowledge gap.


## 4.4 Limitations of deductive procedure used

Despite the varied methods of assessment and wide variety of evidence considered, there are nonetheless clear limitations in the deductive procedures adopted here. Firstly, it is possible that mechanisms for changes in the rainfall-runoff relationships have been missed by the workshop participants, or that we are not aware of the importance of certain factors or their 730 interactions (Hunt and Welter, 2010). While this study focuses on identifying individual common or frequently applicable mechanisms, in reality different causes can act together and in different proportions in different landscapes, possibly leading to unexpected outcomes, or to the same mechanism/interaction producing different outcomes in different landscapes. In cases of two competing processes, merely conceptualising interactions (as here) is often not sufficient, but it is necessary to also quantify the interactions (eg. via simulation) to be able to guess at what the outcome might be. Such quantification is a 735 key recommendation as a next step to this work (see recommendations in Section 5).

The assessment methods are each rather blunt because they rely on generalisations and assessment of large-scale patterns in either space, time or both. This contrasts with common field methods for deducing hydrological processes at-site, which may rely on techniques such as hydrogeochemistry, tracers, etc. Such techniques provide individualised insight for a given 740 catchment, in line with the concept of "uniqueness of place" (Beven, 2000); ie., that each catchment is subject to distinctive hydrological processes. Furthermore, the results of such individual studies may be more directly relatable to hydrological processes. Although these benefits are not present in the methods used here, the strength of this work is a large-scale assessment of hydrologic changes and potential drivers. The landscape-scale patterns of hydrological shifts (eg. Figure 2, Saft et al., 2015; Peterson et al. 2021) require corresponding landscape-scale methods of evaluation. Experimental studies are 745 difficult to conduct over large areas and/or large samples of catchments, and it is difficult to generalise regional impacts from individual field studies. In any case, the Millennium Drought is now in the past so it is not possible to conduct experimental studies to directly confirm or refute hydrological hypotheses related to this event.

Nonetheless, the current study may be used as a prelude to planning more extensive field studies (and/or modelling studies) 750 to test predictions of hypotheses in the present time. It may be possible via new experimental studies to gain insights that are transferable backwards in time to past droughts, and this is discussed further in Section 5. In summary, although the methods used here are subject to clear limitations, they are arguably suited to the present context.



## 5. Future research and monitoring needs

As per Figure 1, the key question of this section is *What future research and/or monitoring is required to decide between and/or harmonise the plausible process explanations?* Given the dominance of complementary HPEs (Section 3.5), it seems unnecessary to "decide between" those remaining (since they can occur together), although it is still desirable to investigate and confirm their relevance. The set of plausible process explanations is dominated by subsurface processes, so an overarching theme is that we need to better understand the role of groundwater directly and indirectly in streamflow generation, including indirect interactions with plant water use and the effect of multi-year declines in groundwater level. It follows that the main monitoring need is for long-term (10+ year) monitoring programs of groundwater and the vadose zone. Given the potential for interaction among processes, it is important to have coordinated monitoring of groundwater, vadose zone, ET fluxes, streamflow, and vegetation conditions to accelerate a holistic understanding of process interactions.

Most bores in south-east Australia have been targeted where groundwater is used, whereas this work affirms the need for investment in bores for hydrological understanding rather than impact assessment. By considering the range of hypotheses being tested, and the range of data sources currently available (including their spatial distributions and uncertainties), it should be possible to pinpoint locations where new bores would yield the greatest advances in understanding. In general, the largest spatial gaps in bore coverage are in upland areas, so this should be considered when planning new monitoring.

For specific HPEs, future research and/or monitoring needs to provide additional evidence and improve confidence include:

- **HPE18 ("Gaining to losing"):** Priority should be placed on retaining and commissioning bores adjacent to streams and rivers. There would likely be considerable value in reviving, even for a short time, decommissioned bores with historic pre-drought data. As with new bores, the focus for recommissioning should be on objectively identifying bores that might provide the most information, complementing existing data and considering uncertainties.

- **HPE16 ("Sustained hillslope drainage"):** This HPE differentiates dynamics in colluvial/alluvial valley bottoms from dynamics in fractured rock in hillslopes, so future research should aim to discern these differentiated dynamics in measurements. It also explains groundwater loss primarily via AET from valley bottoms, and it may be possible to design experiments to detect this using remote sensing or high resolution products.

- **HPE17 ("Sustained regional drainage"):** Further research into the volumetric rates of flow through regional systems is required, for example through analysing water age (eg. Markovich et al., 2019). Most current monitoring of regional systems is done via analysis of levels; quantification of flow rates is relatively rare and would provide valuable insights (even if estimating temporal changes in this rate proves infeasible).

- **HPE19 ("Vadose zone available capacity"):** Simulation of amplification effects may be the most effective test to increase confidence in this hypothesis. For example, if a temporally variable vadose zone thickness were to improve the realism of model simulations of the Millennium Drought, this could be interpreted as support for the HPE. Few widely-used models currently support such dynamics.





Moving away from groundwater to consider other perceptual models:

- **HPE23 ("Farm dams"):** Better integration of datasets and modelling is required. Estimates of farm dam growth have recently been produced (Malerba et al., 2021), and the next step is to use such estimates in modelling (eg. Fowler et al.,
2015; Morden et al., 2021) to quantify additional impact during the Millennium Drought relative to earlier droughts.

- **HPE13 ("Salinity recovery"):** Further confirmation may be difficult since it is difficult to retrospectively identify areas formerly affected by salinity. The most promising avenue may be to collate and standardise existing information from the 1980s and 1990s (some of which may exist non-electronically).

- **HPE04 ("Evaporative demand")** and **HPE07 ("Coupling and radiation/ temperature"):** Simulation experiments
may be used to estimate the water balance impacts of mechanisms. This would require a framework capable of representing the coupling effects (eg. Clark et al., 2015).

- **HPE21 ("Agricultural practices"):** Knowledge gaps exist regarding the prevalence of land management trends. These gaps can be filled via a mixture of literature review (including grey literature) and workshopping with land managers.

The Millennium Drought was longer than other droughts post-1950 (HPE05 "Long Drought") and this point underlies many findings in this paper. However, two pre-1950s droughts - the Federation Drought and World War 2 drought - are more comparable to the Millennium Drought than droughts in the 1950-1990 period (see Verdon-Kidd and Kiem, 2009 for more information on these droughts). Thus, a potential research direction might be to collate historical information, both qualitative and quantitative, about the hydrological impacts of these droughts, to provide new perspectives on whether the
Millennium Drought was unprecedented from a hydrological perspective. Lastly, efforts to gather, homogenise and publicly release historical experimental data relevant to the Millennium Drought may also be of considerable value, and we feel this should be considered a priority for Australian hydrologists. This could include identifying previously unpublished data held by individual academics or research institutions, and/or making connections between previously disparate datasets.

## 6. Conclusion

The Millennium Drought is an important historic case study of a multi-annual event resulting in persistent shifts in the relationship between rainfall and runoff. This paper, being the culmination of a multi-disciplinary eWorkshop, has provided new perspectives on the processes responsible for these hydrological changes, while at the same time demonstrating the value of using shared perceptual models and evaluation as a strategy for community research.

Through community collaboration, we have presented a broad range of hypotheses across categories including climatic forcing, vegetation, soil moisture dynamics, groundwater, and anthropogenic influence. Assessment of these hypotheses against a large body of evidence reveals that many of the hypotheses are not supported by available data. The majority of remaining hypotheses point to the central role of subsurface processes either directly causing the flow reductions or



mediating them via long-memory processes (eg. multi-annual groundwater storage trends). Recommendations for future
research focus on developing monitoring and modelling to further confirm these hypotheses and assess their interactions.

The resulting contribution is threefold:

1. A systematic catalogue of possible mechanisms explaining flow reductions which may be relevant to droughts in
   other times or places and/or to future changing climate. This is the case even for process explanations subsequently
determined to be inconsistent with evidence for this case study.

2. A holistic description of catchment response during the Millennium Drought, identifying numerous mechanisms
   which together can explain the unexpected flow reductions.

3. Multiple recommendations towards better understanding of multi-year climate drying and its hydrological impact,
   which is essential to provide society with the means to anticipate future water crises globally.

**Appendix A: Data sources for Figures 7 and 8**

**Table A1: Data sources and notes for Figures 7 and 8. All weblinks were last accessed 12ᵗʰ April 2022.**

| Figure | Data | Source / notes |
|---|---|---|
| 7a | Elevation | Commonwealth Government (Geoscience Australia) 3-second SRTM-derived DEM: https://data.gov.au/dataset/ds-ga-a05f7892-ef04-7506-e044-00144fdd4fa6 |
| 7a | Water courses | Hydrosheds river dataset, 30s resolution: https://www.hydrosheds.org/ |
| 7b | Mean annual precipitation | Bureau of Meteorology 30-year (1981-2020) average precipitation: http://www.bom.gov.au/jsp/ncc/climate_averages/rainfall/index.jsp (general site) http://www.bom.gov.au/web01/ncc/www/climatology/rainfall/rainan.zip (specific product) |
| 7c | Land cover | Commonwealth Government (Geoscience Australia) Dynamic Land Cover Dataset Version 2.1 http://pid.geoscience.gov.au/dataset/ga/83868 |
| 7d | Groundwater systems | Victorian Government Groundwater Catchment Systems boundaries. This dataset was supplied by the Victorian Government and is not publicly available, although DSE (2012; available at the link below), outlines the derivation (see Sect. 4.2; also Fig. 2), displaying the layer itself in Fig. 31: http://beta.vvg.org.au/maynard/view_resource.php?resource_id=4336&account=a020b5775c3d4447295a4c758a5394ca |
| 8a | Wildfire extents | Victorian Government fire history overlay of most recent fires. Available at: https://discover.data.vic.gov.au/dataset/fire-history-overlay-of-most-recent-fires |
| 8b | Salinity provinces | Victorian Government salinity provinces. The attribute mapped is PCT_AR_SAL (percentage of area salinised). Available at: https://discover.data.vic.gov.au/dataset/salinity-provinces-of-victoria |
| 8c | Groundwater usage | This dataset is not publicly available. The Victorian Government provided the lead author with their dataset of bores for the state of Victoria (N>15,000), including entitlement volumes. This data was post-processed using ArcMap to sum the entitlement volumes across each 0.05° x 0.05° grid cell, then normalised by streamflow from Weligamage et al. (2021; see two rows down). |
| 8d | Farm dams | Used a similar process to the previous row applied to the sum of two publicly available datasets: https://discover.data.vic.gov.au/dataset/farm-dam-boundaries https://discover.data.vic.gov.au/dataset/farm-dam-points-wms |
| 8c & 8d | Interpolated streamflow | Spatially interpolated streamflow (based on observations) from Weligamage et al. (2021). Paper: https://doi.org/10.36334/modsim.2021.k11.weligamage. Download dataset: https://doi.org/10.5281/zenodo.5454798 |
| 7 & 8 | Catchment boundaries, including hydrologic shifts | Saft et al., under review (see references). |





**Appendix B: Drivers of potential spring evapotranspiration (additional figures)**

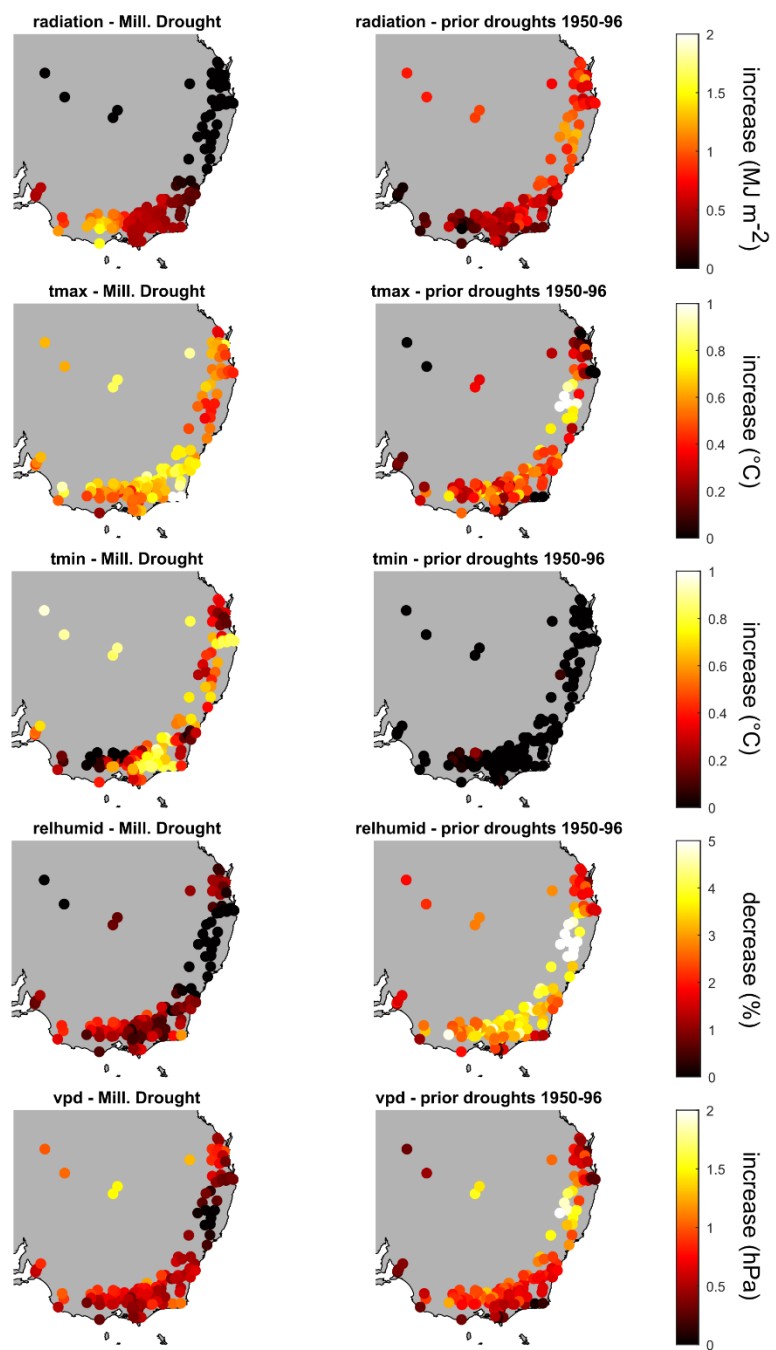

**Figure B1: Repeat of Figure 5d but for five climatic drivers of PET, considering spring months only (September-November). Spring is selected since HPE04 / Figure 5d is focussed on PET in spring. Based on data from Fowler et al. (2021).**





**Figure B2: Repeat of Figure B1 but showing changes in percentage terms rather than actual values.**





**Appendix C: Comparison of climatic conditions: during versus after the Millennium Drought**

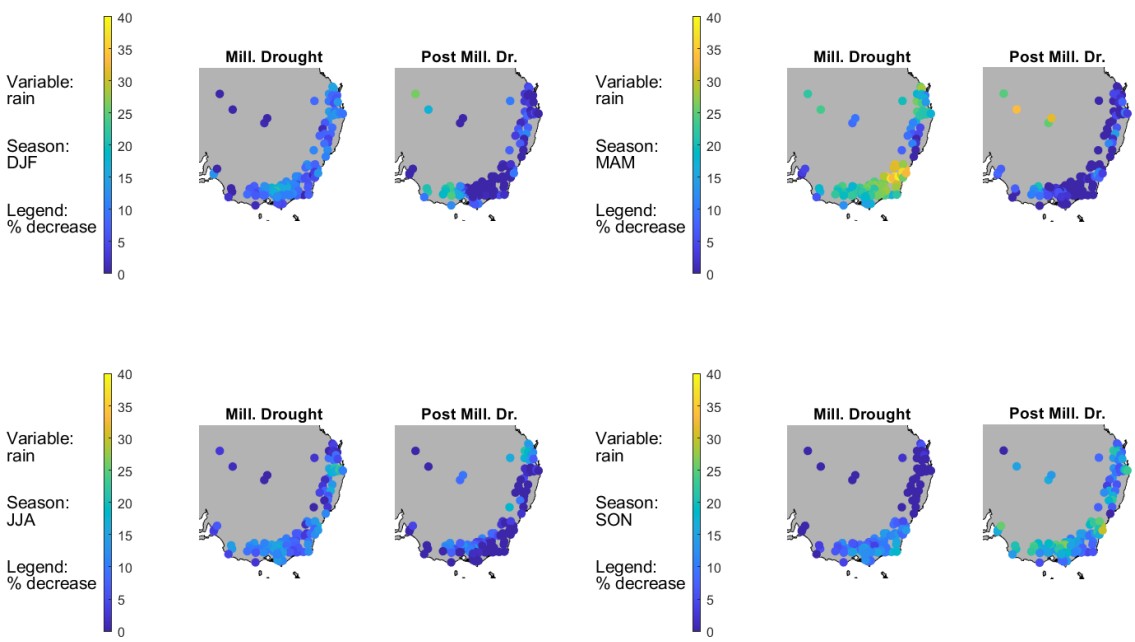

**Figure C1: Comparison for each season showing precipitation. For each pair, the left plot shows the change in Millennium Drought precipitation relative to the long term average (taken as 1950–2017, the same definition as in Figure 3); and the right plot shows the change in post-Millennium-Drought (2011-2019) precipitation relative to the long term average. Based on data from**
**Fowler et al. (2021).**

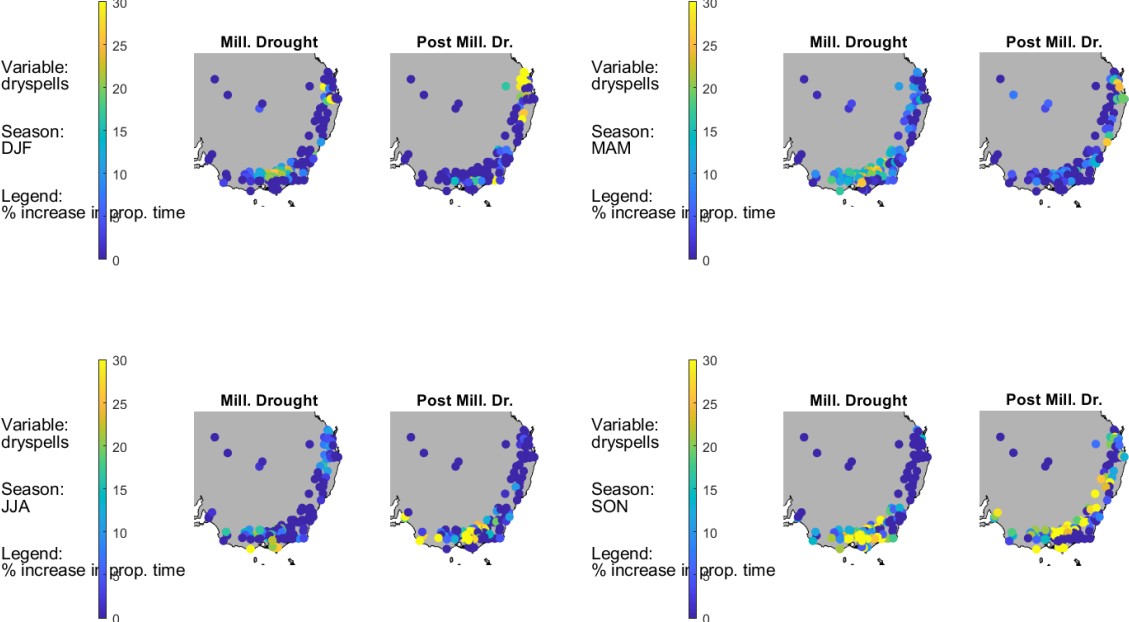

**Figure C2: Same as Figure C1, except for dry spells (see Figure 3 for dry spell definition). Based on data from Fowler et al. (2021).**



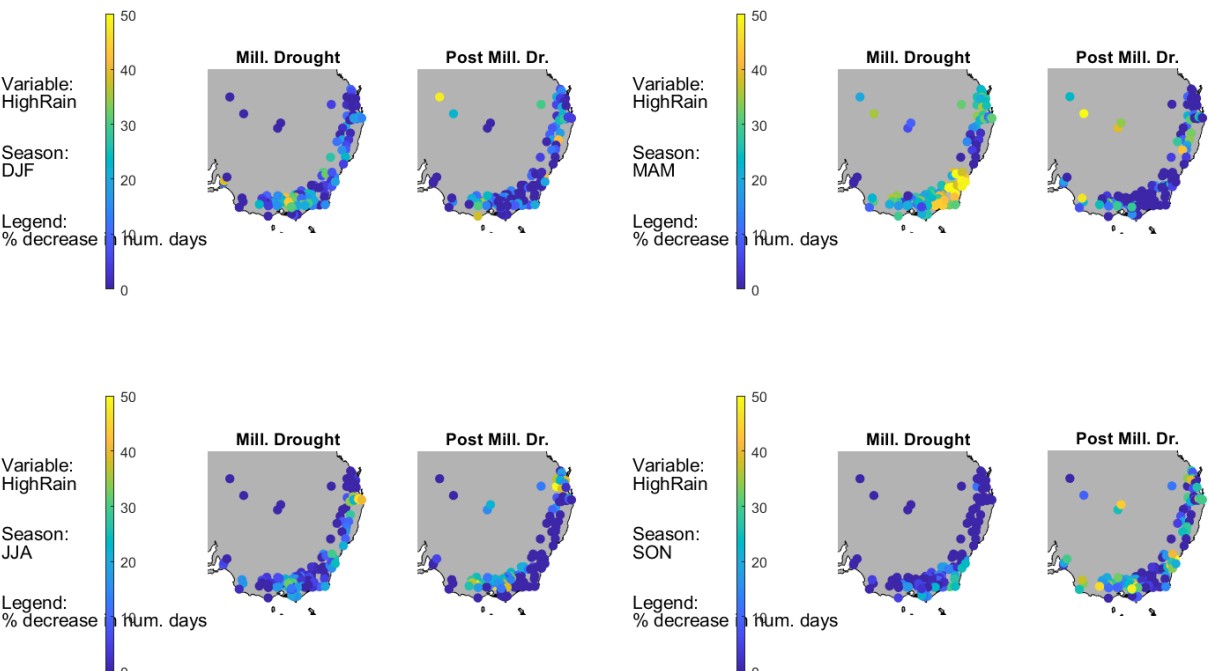

**Figure C3: Same as Figure C1, except for number of high rainfall events (see Figure 3 for high rainfall definition). Based on data from Fowler et al. (2021).**

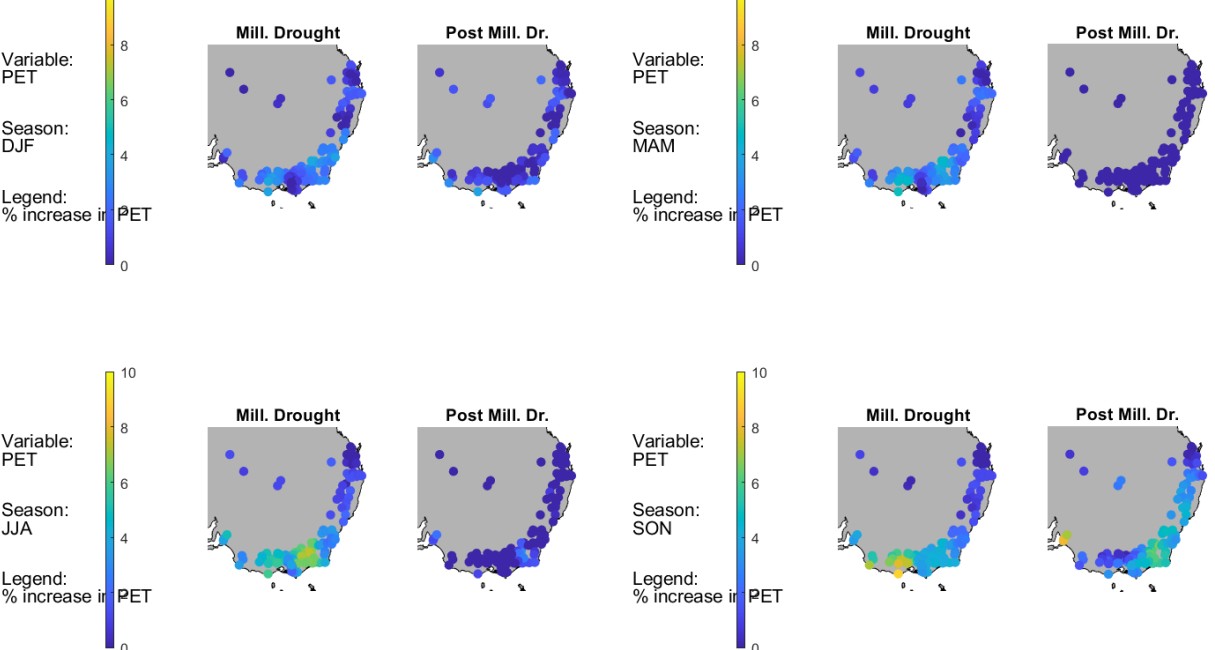

**Figure C4: Same as Figure C1, except for PET (see Figure 3 for PET definition). Based on data from Fowler et al. (2021).**



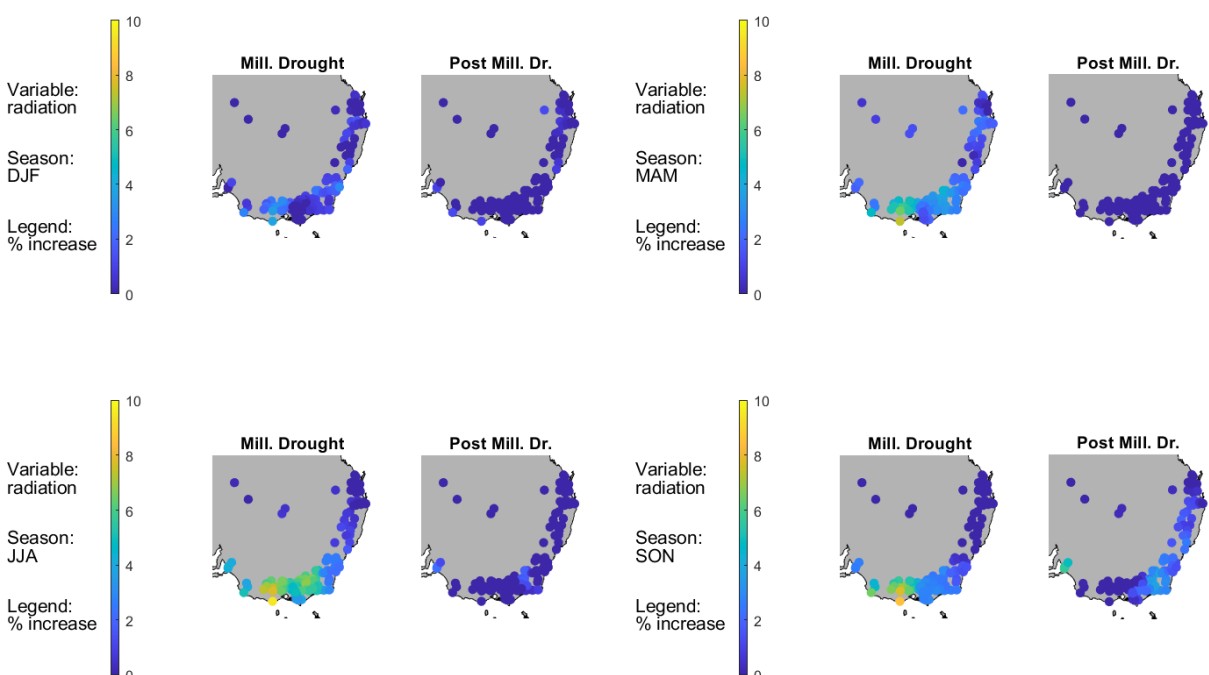

**Figure C5: Same as Figure C1, except for radiation. Based on data from Fowler et al. (2021).**


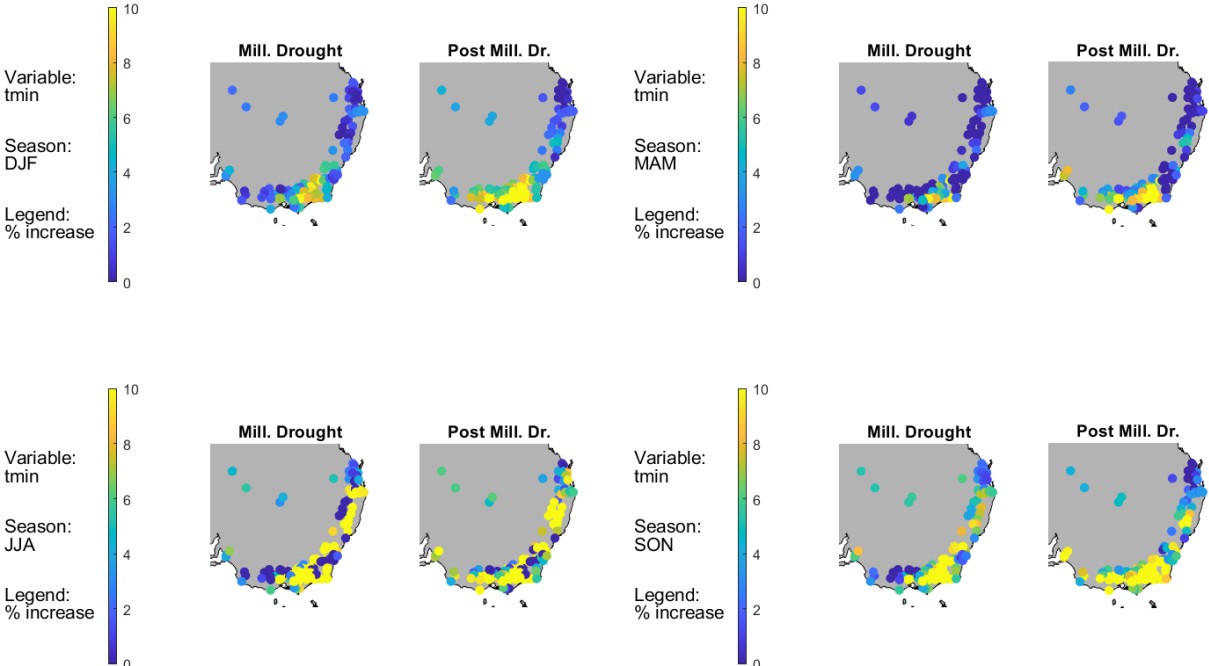

**Figure C6: Same as Figure C1, except for t$_{min}$ (minimum daily temperature). Based on data from Fowler et al. (2021).**



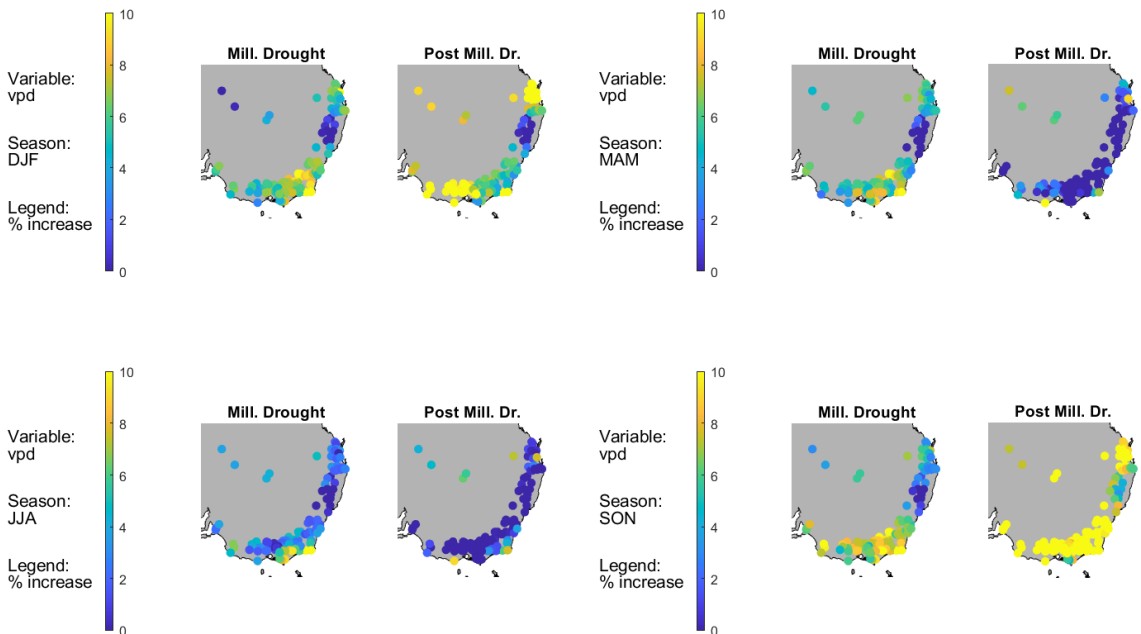

**Figure C7: Same as Figure C1, except for VPD. Based on data from Fowler et al. (2021).**

**Appendix D: Miscellaneous figures**

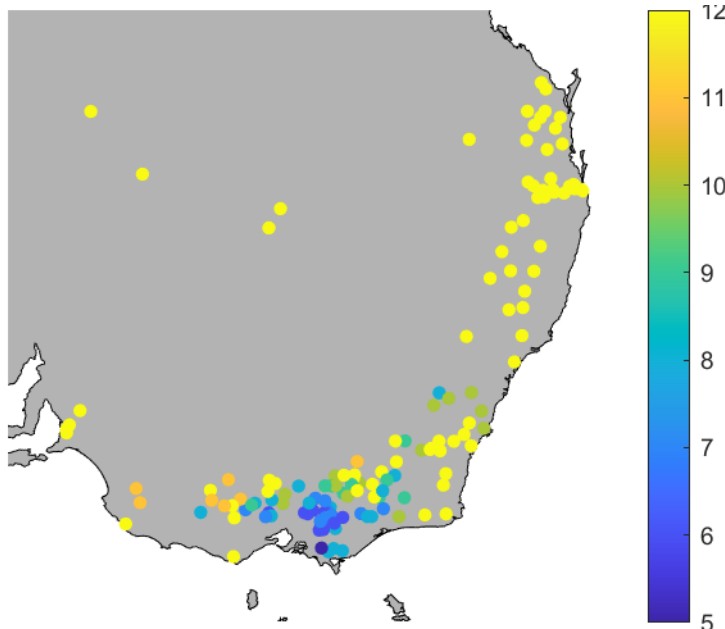

**Figure D1: Number of years that met the definition of "prior droughts" used to create Figure 6, for each catchment.
Based on data from Fowler et al. (2021).**





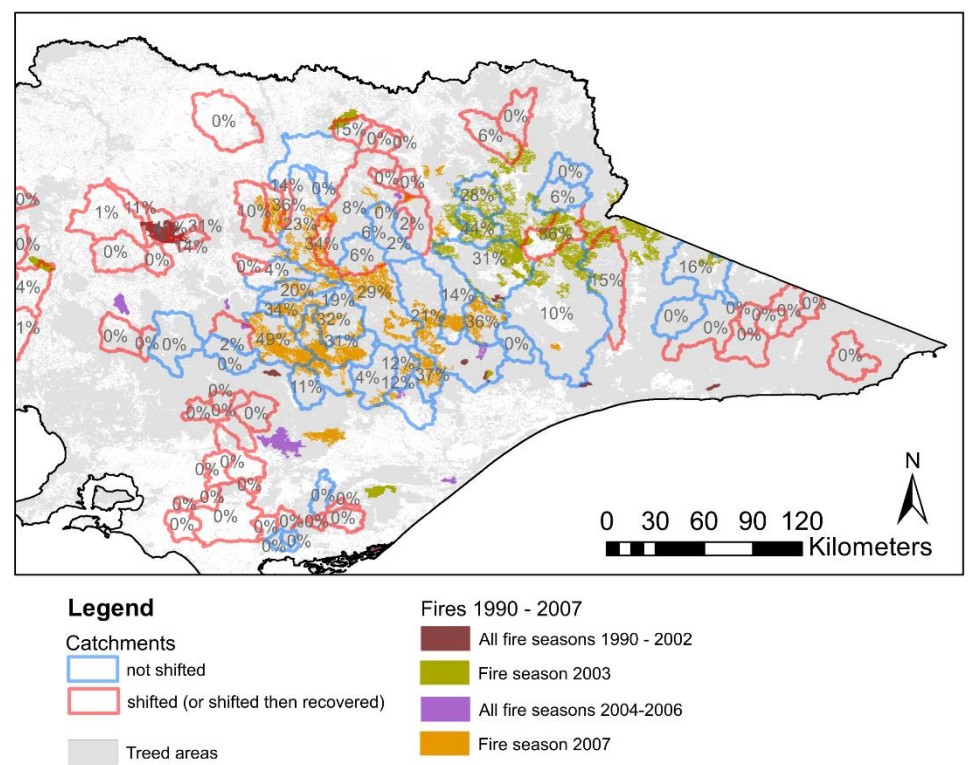

**Figure D2: Fire history (most recent fire) over eastern Victoria, mapped with study catchments from Peterson et al. (2021).**

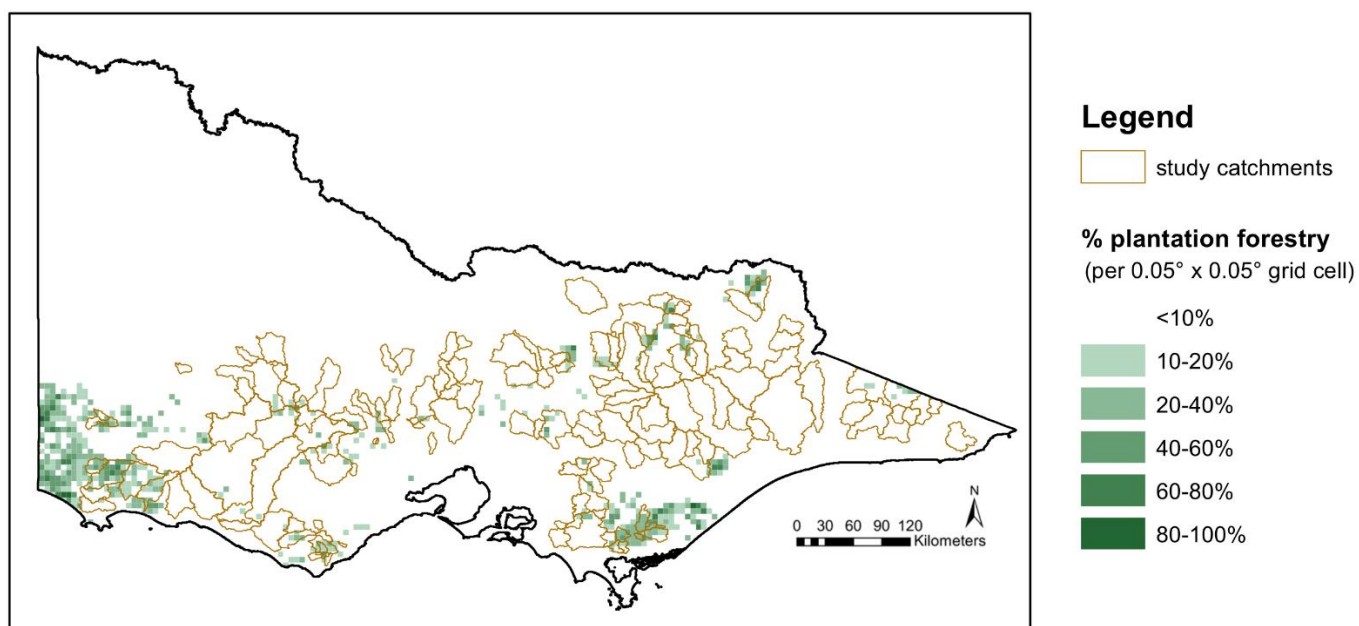

**Figure D3: Spatial distribution of plantation forestry in Victoria in 2016. Processed from raw data from ABARES (2016).**

**Data availability**

All hydroclimatic data used in this study are freely available as part of the CAMELS-AUS dataset (Fowler et al., 2021) at https://doi.org/10.1594/PANGAEA.921850. Non hydroclimatic data sources are marked in each figure (see also Table A1).

**Author contributions**

KF conceptualised this article/workshop and acted as lead convener for the workshop. MP, MS and TP co-convened the workshop, while RN and KF facilitated. All other authors contributed equally to the workshop and to this article. KF coordinated the writing of the article, with input from all co-authors.

**Competing interests**

Some authors are members of the editorial board of HESS. The peer-review process was guided by an independent editor, and the authors have no other competing interests to declare.

**Acknowledgements**

The workshop and article preparation were conducted as part of Australian Research Council (ARC) Linkage Project LP180100796, supported by the Victorian Department of Land, Environment, Water and Planning (DELWP) and Melbourne Water. The same linkage project supported the research of M. Saft, K. Fowler, T. Peterson and M. Peel. K. Fowler also received support from LP170100598 supported by the ARC, DELWP, the Victorian Environmental Water Holder and Australia's Bureau of Meteorology (BoM). A. Ukkola was supported by the ARC Centre of Excellence for Climate Extremes (CE170100023) and an ARC Discovery Early Career Researcher Award (DE200100086). C. Wasko is supported by an ARC Discovery Early Career Researcher Award (DE210100479). E. Daly and I. Cartwright were supported by the ARC Discovery Project DP180101229. G. Bonotto was supported by the Melbourne Research Scholarship, the Victorian Department of Environment, Land, Water & Planning, and Goulburn Broken Catchment Management Authority (TP 707158). AS Kiem was supported by ARC Discovery Project (DP180102522, "Flooding in Australia—are we properly prepared for how bad it can get?". The authors thank Dr Wouter Knoben for critical feedback on the draft manuscript.

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
