# Peer review of "Explaining changes in rainfall-runoff relationships during and after Australia's Millennium Drought: a community perspective"

_Hydrology and Earth System Sciences, 2022_

## Referee Comment (RC2)

In this manuscript the authors analyse the effects of the Millennium Drought with the aim of identifying possible process explanations for the observed persistent changes in the hydrological response in many catchments in Australia.

The experiment was logically designed and very systematically implemented. In a concerted and probably unique effort, the authors formulated an exhaustive suite of potential processes hypotheses. These hypotheses have then been rigorously confronted and scrutinized with observations. The extremely well documented analysis together with the very robust data support and the detailed and critical interpretation thereof make this manuscript an excellent example of good and relevant science. I commend the authors for this effort.

The manuscript is also very clearly structured and well-written. Overall, I do only have two major observations/comments the authors may want to consider:

(1) Although I highly appreciate the detailed explanations of the individual process hypotheses, some of the hypotheses could strongly benefit from a more precise terminology and/or clearer description. This will make it easier for the reader to appreciate and understand the actual differences between different hypotheses (see below in the list of detailed comments).

(2) Some of the hypotheses could benefit from a stronger and wider connection to literature, in particular outside Australia by providing more references to related studies (please note: below I have added a few suggestions. However, these include for my convenience and to save time, quite some work from our group. Please do *not* feel obliged to cite these papers - other research groups may have published material that fits better).

Detailed comments:

P.2, l.50: not sure that "recently" is the most suitable term here. Literature dedicated to the topic has been around for a while. For example, Destouni et al. (2013), Jaramillo and Destouni (2014) or van der Velde et al. (2014) were published almost a decade ago.

P.2, l.52: not sure China qualifies as a "continent". Perhaps worth to also include the recent analysis by Roodari et al. (2021) in Central Asia here.

P.3, l.71-75: there are quite some ongoing initiatives to address this issues and to find work-arounds. Some recent examples include Speich et al. (2020) and Bouaziz et al. (2022).

P.3, l.81: Roodari et al. (2021) in Central Asia

P.5, l.149-151, Figure 1: Excellent approach and description!

P.6, 171-172: Please define how "shift" was defined here. How much of a change is necessary to be considered as "shift"?

P.6, l.177: "persisting within a low runoff state" sounds awkward, given that runoff is a *change of a state* (i.e. dS/dt!). I know what you intend to say but please try to rephrase.

P.6, l.179-180: storages (or states – interception, groundwater storage) are lumped with changes of storages (dS/dt, i.e fluxes – precipitation, ET, recharge). Please avoid that as they have fundamentally different functions. In addition, it is not clear what "ET" stands for here, potential evaporation or actual evaporation? I cannot fully follow the reasoning: increases in actual evaporation are in many cases the direct consequence of increased interception (e.g. on canopies). Thus, they come hand in hand. Please clarify.

P.7, Figure 2: the reader can only assume that in panel (a) the bar chart indicates precipitation and the cross/lines stream flow. Please explicitly describe that in the caption. For panels (c) and (d) please avoid using red and green shades in the same figure: ~15% of your readers will be red-green colour blind.

P.8, l.195: is this so? Then these studies may be based on a somewhat incomplete definition of "drought". In any case, please provide supporting reference for this statement or remove it.

P.8, l.200-202: this is a repetition of P.5, L.135-139 and Figure 1. Can be omitted.

P.8, l.218ff: although the explanation of the "two water world" (TWW) hypothesis is correct here, it does not at all support your argument here. TWW describes actual water ages and the related transit and residence time distributions that are largely controlled by the physical transport *velocities* of individual water molecules. Here, seasonal or annual water budgets are considered. These are instead controlled by *response times* which are regulated by the propagation of pressure waves at given *celerities* (see McDonnell and Beven, 2014; and Figure 2a in Hrachowitz et al., 2016). The reference to the TWW is therefore unsuitable and actually incorrect here. Instead what you describe here ("…water can move to the stream only after soil pores have been replenished") is the functioning and role of water storage following the concepts of Field Capacity and Permanent Wilting Point. Please remove any reference to TWW here.

P.11, l.240 (also Table 1): not sure how HPE02 is different to HPE02. Both are effectively the result of higher water deficits (i.e. lower water content below Field capacity) due to more pronounced dry periods.

In both cases, the water deficit is not overcome – water content in soils remains below field capacity and is therefore held against gravity instead of being released directly (or via groundwater) to the stream. Please clarify the difference between HPE01 and HPE02. In addition, please specify more precisely what is meant by "initial losses". Do you mean water stored in soils and eventually released as transpiration or (to a minor degree) soil evaporation? Where else could this water be lost to?

P.12, l.270: "activation of long-term…processes" is very vague. Please try to be more specific. How does the reader have to imagine that? Is it really a binary phenomenon (active vs. deactivated?) or is it a process that gradually and proportionally becomes more relevant and visible the drier the system becomes?

P.12, l.279: please avoid absolute terms such as "veracity". The best we can do in large-scale hydrology is to test and evaluate our hypothesis.

P.12, l.293ff: perhaps good to refer to Jaramillo et al. (2018), who provide a good illustration of the counteracting effects of fertilization.

P.13, l.304: it is not quite clear to me (1) why HPE07 falls under "vegetation conditions" and not under "meteorological dynamics", (2) what the difference between HPE07 and HPE04 is, as both radiation and temperature are major controls on evaporative demand (HPE04, i.e. potential evaporation) and (3) why in the one sentence description the focus of HPE07 is radiation and temperature, while in the text just above, the turbulence differences are the actual core of this HPE. Please clarify.

P.13, l.322: agreed, but they will not only intercept water on their foliage and thus allow evaporation, but they will also continue to transpire water. Please try to be more specific here.

P.13, l.324-324: perhaps explicitly refer to C4 grass here.

P.14, l.331: why only in south-east Australia? I would suspect this to be a general phenomenon.

P.14, l.336: not sure what is meant by "assuming some other mechanism for lower stream flow,…". Please clarify and rephrase.

P.14, l.344: see above

P.15, l.362: see above

P.15, l.365ff: Not clear what the actual processes involved here are meant to be. In my understanding the idea is that upwelling groundwater, persistent over several decades dissolved salts from deeper parts of the soil and moved it to near-surface layers. If this is so, I do not understand how lowering the groundwater tables would reduce salinity over a relatively short period: solute movement in the subsurface, in particular during dry conditions (i.e. droughts!) is characterized by very slow transport velocities and long-term legacy effects (e.g. Basu et al., 2010; Hrachowitz et al., 2015). Instead, solutes moved into near-surface layers during wet conditions will frequently undergo evapo-concentration effects (e.g. Hrachowitz et al., 2015), thus temporarily making conditions for plant growth even more unfavourable.

P.15, l.376: the use of the term "interception" here and elsewhere in the manuscript is ambiguous. Please note that "interception" has a very specific meaning in hydrological literature (e.g. Miralles et al., 2020; Savenije, 2004). What is specifically meant here? A process that retains water on the canopy, foliage or near-surface soil layers to supply water for the "evaporation" process or is it used in a more general way to also include "interception" in in the root zone to supply root-water uptake for transpiration? If it is the latter, I strongly recommend to rephrase to avoid misunderstandings.

P.15, 377ff: HPE15 is described only in very broad and vague terms. I agree, that the actual processes here may be unknown. However, in such a case, I am not sure if a meaningful hypothesis can be formulated, because a meaningful hypothesis always needs to be testable, otherwise it cannot be qualified as hypothesis. Please try to more explicitly specify this hypothesis or remove it, as it may be indistinguishable from most of the other vegetation-related HPEs here.

P.16, l.392: not sure if "activated" is the most suitable term here (see one of the comments above)

P.16, l.403: what is "diffuse discharge"?

P.16, l.411: please specify "discharge areas"

P.16, l.414: "interception by transpiration"? Please see comment above.

P.18, l.435: Hulsman et al. (2021a) similarly found supporting evidence for the importance of upland groundwater sustaining alluvial evaporation/transpiration in a large scale study in the Zambezi basin. Perhaps nice to include as reference.

P.18, l.444-446: explicitly mention the role of evaporation/transpiration here to be more specific.

P.18, l.449: potentially very relevant and ofter overlooked process. Bouaziz et al. (2018), Condon et al. (2020), Hulsman et al. (2021b) but also the authors themselves (Fowler et al., 2020) provide different recent perspectives on the potential importance of this process. Frisbee et al. (2012) also provide an excellent synthesis and illustration (Figure 3 therein!). Would be good to add at least some of these references here to provide a stronger context and background for the reader.

P.19, l.461: "[…] some systems […] that are more extensive […] have longer response times […]". Really? I am not sure this reasoning is generally valid. In larger systems the average flow distance in the subsurface domain, the controlling factor on response times at time scales > one month, is in most environments very similar to those of smaller, headwater systems. In other words, no matter if you are anywhere upstream or downstream in the landscape, the distance to the next river will not be that different, due to the fractal, scale-invariant nature of river networks (e.g. Rodriguez-Iturbe and Rinaldo, 2001). Please provide a reference or remove.

P.19, l.485ff: this HPE needs more explanation. It is not clear why the vadose zone should indeed be drier. The reasoning here, as far as I understand, is that declining GW tables result in deeper vadose zones. Ok. In these parts of the subsurface, all the water that cannot be held against gravity (water above Field Capacity) will released and "follow" the falling GW table. The remainder, i.e. soil water content at Field Capacity, will largely be held against gravity. This water can only be released by evaporation or plant water uptake for transpiration. Assuming that in many locations the GW-table is below the root zone, plant water uptake drops out as potential process to remove water. However, soil water at depth below 20-30cm can also not be evaporated at very high rates (e.g. Brutsaert, 2014). The deeper, the less relevant soil evaporation will be due to the limited diffusive gas/vapour exchange with the surface (there is no wind in the soil pores for turbulent exchange!). Soil deeper below the root zone will thus frequently be close to Field Capacity, as the water cannot be released with gravity only and very limited evaporation. It would be great if you could provide a more detailed description of your hypothesis that soils in a deeper vadose zone can be drier (i.e. below Field Capacity – because if they remain at or above field capacity, they will be hydraulically and hydrologically irrelevant. In that case, all the water that enters this zone from above will be again released as it cannot be held, i.e. dS/dt ~ 0 over time scales larger than a few days).

P.19, l.487-488: I suspect you mean that the "infiltration capacity" declines with increasing wetness as described by Darcy-Richards. In contrast, "hydraulic conductivity" typically increases with increasing wetness!

P.20, l.495: that cracked soils result in less runoff is of course not impossible. However, also the opposite, the importance of cracks as preferential flow pathways, is frequently observed and documented (e.g. Zehe at al., 2013). Please adjust the hypothesis accordingly.

P.20, l.497-498: "Streamflow was lower […] because […] higher infiltration […]". This does not quite add up for me. Water that infiltrates surely does not disappear. Was the assumption that most of it will be held in the soil and evaporated/transpired instead of recharging the GW?

P.21, l.529: perhaps better to replace "interception" by something like "retention and subsequent evaporation/transpiration"

P.23, l.584: should this read as "$10^5$"?

P.23, l.600: please also explicitly mention the three plausible hypotheses here and not only in the table.

P.24, HPE08: "[…] it is doubtful whether modest historical $CO_2$ increases could have caused larger changes […]". Without any further data support, more detailed reasoning and/or references this remains largely speculation and cannot be used as hypothesis test.

P.25, HPE13: in the light of my comments above, the reasoning here (low water table) is not very convincing.

P.26, HPE19: see above. A deeper vadose zone will only allow further water retention if water from pores is being extracted by soil evaporation and or transpiration. Otherwise the zone will, on average, remain close to Field Capacity and act as a hydraulically and hydrologically passive part of the soil. In other words, it will cause some delay in the water percolating through this zone, but it will not provide additional "storage", i.e. on time scales of more than a few days dS/dt~0.

P.33, 684ff: see above. Also, the term "interception" is not suitable here

References:

  Basu, N. B., Destouni, G., Jawitz, J. W., Thompson, S. E., Loukinova, N. V., Darracq, A., ... & Rao, P. S. C. (2010). Nutrient loads exported from managed catchments reveal emergent biogeochemical stationarity. Geophysical Research Letters, 37(23).

Bouaziz, L., Weerts, A., Schellekens, J., Sprokkereef, E., Stam, J., Savenije, H., & Hrachowitz, M. (2018). Redressing the balance: quantifying net intercatchment groundwater flows. Hydrology and Earth System Sciences, 22(12), 6415-6434.

Bouaziz, L. J., Aalbers, E. E., Weerts, A. H., Hegnauer, M., Buiteveld, H., Lammersen, R., ... & Hrachowitz, M. (2022). Ecosystem adaptation to climate change: the sensitivity of hydrological predictions to time-dynamic model parameters. Hydrology and Earth System Sciences, 26(5), 1295-1318.

Brutsaert, W. (2014). Daily evaporation from drying soil: Universal parameterization with similarity. Water Resources Research, 50(4), 3206-3215.

Condon, L. E., Markovich, K. H., Kelleher, C. A., McDonnell, J. J., Ferguson, G., & McIntosh, J. C. (2020). Where is the bottom of a watershed?. Water Resources Research, 56(3), e2019WR026010.

Destouni, G., Jaramillo, F., & Prieto, C. (2013). Hydroclimatic shifts driven by human water use for food and energy production. Nature climate change, 3(3), 213-217.

Fowler, K., Knoben, W., Peel, M., Peterson, T., Ryu, D., Saft, M., ... & Western, A. (2020). Many commonly used rainfall-runoff models lack long, slow dynamics: Implications for runoff projections. Water Resources Research, 56(5), e2019WR025286.

Frisbee, M. D., Phillips, F. M., Weissmann, G. S., Brooks, P. D., Wilson, J. L., Campbell, A. R., & Liu, F. (2012). Unraveling the mysteries of the large watershed black box: Implications for the streamflow response to climate and landscape perturbations. Geophysical Research Letters, 39(1).

Hrachowitz, M., Fovet, O., Ruiz, L., & Savenije, H. H. (2015). Transit time distributions, legacy contamination and variability in biogeochemical $1/f\alpha$ scaling: how are hydrological response dynamics linked to water quality at the catchment scale?. Hydrological Processes, 29(25), 5241-5256.

Hrachowitz, M., Benettin, P., Van Breukelen, B. M., Fovet, O., Howden, N. J., Ruiz, L., ... & Wade, A. J. (2016). Transit times—The link between hydrology and water quality at the catchment scale. Wiley Interdisciplinary Reviews: Water, 3(5), 629-657.

Hulsman, P., Savenije, H. H., & Hrachowitz, M. (2021a). Learning from satellite observations: increased understanding of catchment processes through stepwise model improvement. Hydrology and Earth System Sciences, 25(2), 957-982.

Hulsman, P., Hrachowitz, M., & Savenije, H. H. (2021b). Improving the representation of long-term storage variations with conceptual hydrological models in data-scarce regions. Water Resources Research, 57(4), e2020WR028837.

Jaramillo, F., & Destouni, G. (2014). Developing water change spectra and distinguishing change drivers worldwide. Geophysical Research Letters, 41(23), 8377-8386.

Jaramillo, F., Cory, N., Arheimer, B., Laudon, H., Van Der Velde, Y., Hasper, T. B., ... & Uddling, J. (2018). Dominant effect of increasing forest biomass on evapotranspiration: interpretations of movement in Budyko space. Hydrology and Earth System Sciences, 22(1), 567-580.

McDonnell, J. J., & Beven, K. (2014). Debates—The future of hydrological sciences: A (common) path forward? A call to action aimed at understanding velocities, celerities and residence time distributions of the headwater hydrograph. Water Resources Research, 50(6), 5342-5350.

Miralles, D. G., Brutsaert, W., Dolman, A. J., & Gash, J. H. (2020). On the use of the term "evapotranspiration". Water Resources Research, 56(11), e2020WR028055.

Roodari, A., Hrachowitz, M., Hassanpour, F., & Yaghoobzadeh, M. (2021). Signatures of human intervention–or not? Downstream intensification of hydrological drought along a large Central Asian river: the individual roles of climate variability and land use change. Hydrology and Earth System Sciences, 25(4), 1943-1967.

Savenije, H. H. (2004). The importance of interception and why we should delete the term evapotranspiration from our vocabulary. Hydrological processes, 18(8), 1507-1511.

Speich, M. J., Zappa, M., Scherstjanoi, M., & Lischke, H. (2020). FORests and HYdrology under Climate Change in Switzerland v1. 0: a spatially distributed model combining hydrology and forest dynamics. Geoscientific Model Development, 13(2), 537-564.

Van der Velde, Y., Vercauteren, N., Jaramillo, F., Dekker, S. C., Destouni, G., & Lyon, S. W. (2014). Exploring hydroclimatic change disparity via the Budyko framework. Hydrological Processes, 28(13), 4110-4118.

---

## Author Comment (AC1)

| | **Comment**
*(line numbers refer to the original HESS-D submission)* | **Response**
*(line numbers refer to the updated manuscript, non-track-changes version)* |
|---|---|---|
| | *Reviewer 1: Dengfeng Liu* | |
| 1 | General comments: The manuscript presents a range of possible process explanations of flow response to the Millennium Drought in Australia, and then evaluates these hypotheses against available evidence. The manuscript is an excellent work to understand the changes in rainfall-runoff relationships after Australia's Millennium Drought. The strength of this work is a large-scale assessment of hydrologic changes and potential drivers. The framework of Hypothesised Process Explanations is also useful to investigate the effects of the drought in other watersheds, and planning more extensive field studies to test predictions of hypotheses. Specific comments: | Thank you for the supportive comments and constructive feedback. |
| 2 | In Figure 2, the data of precipitation and runoff from 2011 to 2021 should also be presented to show the hydrological behavior after the Millennium Drought. | Agreed that post-drought data is informative. We have added data up until 2017 (Line 193), and we hope this suffices. Data up to 2021 are not easily available because the source dataset (Hydrologic Reference Stations / CAMELS AUS) is not yet updated to 2021. |
| 3 | Line 234, The manuscript focus on the changes of rainfall-runoff relationships, the annnual runoff coefficient, and those in each season may be necessary to discuss. | It is reasonable to provide the reader with information about runoff coefficients (including seasonally) as background information to assist interpretation of the HPEs. As such, we added Appendix B, which provides maps of annual and seasonal runoff coefficients (along with precipitation and streamflow) over the pre-drought, drought and post-drought periods. We added a reference to the new appendix near to where the reviewer commented (Line 168). We did not feel the runoff coefficients needed to be discussed beyond their being mentioned in Section 2 (Line 168) and as part of various HPEs (eg. HPE01, Line 216). |
| 4 | The multiple stable states of the watershed may be a potential perspective to explain the change of the behavior of the rainfall-runoff relationship, as mentioned in Line 379, such as that in Peterson et al. (2009). The Millennium Drought is an extreme disturbance that may push the system from one stable state to another. The question is how to quantify the multiple stable states of the watershed. The dry stable state may be seldom presented in the watershed. Peterson T J, Argent R M, Western A W, Chiew F H S. Multiple stable states in hydrological models: An ecohydrological investigation, Water Resources | These concepts were already present in the manuscript, but perhaps less explicitly. In response to comments from both reviewers, we have expanded the text on this topic. It now reads (line 398):

*"Peterson et al. (2021) ... suggest that their findings (as summarised in Section 2) are "consistent with ... watersheds having multiple stable states [of behaviour] and a finite resilience". In this view, the hydrological shift towards a lower rainfall-runoff relationship corresponds to a transition from one stable state to another (see Peterson et al., 2009). It is noted* |

| | **Comment**
*(line numbers refer to the original HESS-D submission)* | **Response**
*(line numbers refer to the updated manuscript, non-track-changes version)* |
|---|---|---|
| | Research, 2009, 45, W03406, doi:10.1029/2008WR006886. | *that the word "state" here has a different meaning to its common hydrological usage (eg. "state variable"), as it refers to the behaviour of a system being organised into discrete preferential states (eg. the "wet" and "dry" states of D'Odorico and Porporato (2004)), with intermediate conditions having low probability of occurrence. The Millennium Drought was an extreme disturbance that might have pushed several catchments from a wetter to a drier state, with the drier state otherwise seldom being apparent in a catchment's behaviour."* |
| 5 | If the water storage capacity in a watershed is large enough to control all/most of the annual runoff (associated with HPE23), the watershed will be a human-controlled system where the released runoff is regulated by the reservoirs, such as Tarim River basin in China (Liu et al., 2014; Liu et al., 2015). The total water storage capacity of all dams in a watershed may be an important index.Liu, Y., Tian, F., Hu, H., Sivapalan, M. Socio-hydrologic perspectives of the co-evolution of humans and water in the Tarim River basin, Western China: the Taiji–Tire model[J]. Hydrol. Earth Syst. Sci., 2014, 18, 1289-1303. Liu D, Tian F, Lin M, Sivapalan M. A conceptual socio-hydrological model of the co-evolution of humans and water: case study of the Tarim River basin, western China[J]. Hydrology and Earth System Sciences, 2015, 19(2): 1035-1054. | We have added the following text at line 563:

*"Thus, the total water storage capacity of all dams in a watershed may be an important index, particularly when relativised by precipitation or streamflow to account for aridity; that is, a given storage size may have a higher relative impact in drier areas (eg. Liu et al., 2015)".*

Note that we map this index later in the article, in Figure 8d. Also, the Liu et al. (2015) is a very interesting case study, thank you! |
| 6 | Line 570, the spatial distribution of the driving factor should be consistent with the spatial distribution of shifted versus unshifted catchments. Maybe an example will be helpful to understand it. | We have added the following text at line 614:

"*For example, concerning radiation increases during the Millennium Drought, the spatial distribution of these increases (Figure C1) across the southern state of Victoria broadly matches the spatial distribution of shifted catchments, lending credence to HPE07*" |
| 7 | Line 600, Of the twenty-four HPEs, three are considered plausible, ten are considered inconsistent with evidence, and eleven are in a category in-between. The strength of this work is a large-scale assessment of hydrologic changes and potential drivers. This information should be stated in abstract. | This has been added, thank you for the suggestion. (Line 40) |
| 8 | In Figure 10, Higher AET per mm of rainfall, and it equals aridity index=AET/rainfall. | It's possible that there are numerous different definitions of aridity, but we note that most of them involve PET not AET. For example, possibly the earliest aridity index is that of Oldekop in 1911, who used the idea of long-term |

| | Comment
*(line numbers refer to the original HESS-D submission)* | Response
*(line numbers refer to the updated manuscript, non-track-changes version)* |
|---|---|---|
| | | maximum evaporation divided by long-term precipitation to define an aridity index (see http://dx.doi.org/10.1016/j.jhydrol.2016.02.002). Likewise, the PET/P ratio (or similar) underlies most versions of Budyko-type frameworks (eg. https://doi.org/10.1016/j.advwatres.2019.103435) |
| 9 | Technical corrections: | |
| 10 | L227 and L240, event rainfall->rainfall events | We have changed this as requested and also reworded it in response to the other reviewer (line 247). |
| 11 | Line 714, check the citation of Figure 4c. Maybe it is Figure 5d. | Thank you for noticing this error; it has now been corrected. |

---

## Author Comment (AC3)

| | Comment
*(line numbers refer to the original HESS-D submission)* | Response
*(line numbers refer to the updated manuscript, non-track-changes version)* |
|---|---|---|
| | *Reviewer 2: Markus Hrachowitz* | |
| 1 | In this manuscript the authors analyse the effects of the Millennium Drought with the aim of identifying possible process explanations for the observed persistent changes in the hydrological response in many catchments in Australia. | |
| 2 | The experiment was logically designed and very systematically implemented. In a concerted and probably unique effort, the authors formulated an exhaustive suite of potential processes hypotheses. These hypotheses have then been rigorously confronted and scrutinized with observations. The extremely well documented analysis together with the very robust data support and the detailed and critical interpretation thereof make this manuscript an excellent example of good and relevant science. I commend the authors for this effort. | We very much appreciate your encouragement! |
| 3 | The manuscript is also very clearly structured and well-written. Overall, I do only have two major observations/comments the authors may want to consider: | |
| 4 | (1) Although I highly appreciate the detailed explanations of the individual process hypotheses, some of the hypotheses could strongly benefit from a more precise terminology and/or clearer description. This will make it easier for the reader to appreciate and understand the actual differences between different hypotheses (see below in the list of detailed comments). | Thank you. We found the points of clarification to be very helpful to improve the manuscript. |
| 5 | (2) Some of the hypotheses could benefit from a stronger and wider connection to literature, in particular outside Australia by providing more references to related studies (please note: below I have added a few suggestions. However, these include for my convenience and to save time, quite some work from our group. Please do *not* feel obliged to cite these papers - other research groups may have published material that fits better). | Thank you. As a nearly all-Australian authorship, we naturally highlighted Australian studies of the topics. The broadening of the cited literature, as per your suggestions, was most welcome and improved the manuscript. |
| 6 | Detailed comments: | |
| 7 | P.2, l.50: not sure that "recently" is the most suitable term here. Literature dedicated to the topic has been around for a while. For example, Destouni et al. (2013), Jaramillo and Destouni (2014) or van der Velde et al. (2014) were published almost a decade ago. | This is reasonable, and the text now reads (Line 55):

"*[these topics] are receiving increased attention over the last decade or so (eg. Jaramillo and Destouni, 2014; Van der Velde, 2014)*" |

| | Comment
*(line numbers refer to the original HESS-D submission)* | Response
*(line numbers refer to the updated manuscript, non-track-changes version)* |
|---|---|---|
| 8 | P.2, l.52: not sure China qualifies as a "continent". Perhaps worth to also include the recent analysis by Roodari et al. (2021) in Central Asia here. | Thanks, it now reads (Line 57):

"*multiple continents including North America, South America, Europe, Asia and Australia*"

Plus, two recent studies (including the one suggested) are newly cited. |
| 9 | P.3, l.71-75: there are quite some ongoing initiatives to address this issues and to find work-arounds. Some recent examples include Speich et al. (2020) and Bouaziz et al. (2022). | Thank you for these relevant references. They are now cited as examples following the sentence mentioning "transient ecosystem dynamics". (Line 73) |
| 10 | P.3, l.81: Roodari et al. (2021) in Central Asia | Added, thank you. (Line 88) |
| 11 | P.5, l.149-151, Figure 1: Excellent approach and description! | Thank you! |
| 12 | P.6, 171-172: Please define how "shift" was defined here. How much of a change is necessary to be considered as "shift"? | Added the following text (line 175):

"*The degree of change necessary to be considered a 'shift' varied by catchment, since the key criterion was a statistical test designed to test significant differences in relationships (α=0.05) and results of this test are sensitive to background temporal variability.*" |
| 13 | P.6, l.177: "persisting within a low runoff state" sounds awkward, given that runoff is a *change of a state* (i.e. dS/dt!). I know what you intend to say but please try to rephrase. | Apologies, this language derives from Peterson's method and is not necessary here. Changed to (line 116):

"*persisting within a lower rainfall-runoff relationship*".

See also row 32 of this table. |
| 14 | P.6, l.179-180: storages (or states – interception, groundwater storage) are lumped with changes of storages (dS/dt, i.e fluxes – precipitation, ET, recharge). Please avoid that as they have fundamentally different functions. In addition, it is not clear what "ET" stands for here, potential evaporation or actual evaporation? I cannot fully follow the reasoning: increases in actual evaporation are in many cases the direct consequence of increased interception (e.g. on canopies). Thus, they come hand in hand. Please clarify. | Apologies for confusion. We have simplified the sentence so it now reads: "*Peterson et al. (2021) suggested the non-recovery was found to be best explained by increased actual evapotranspiration per unit of precipitation, rather than alternatives such as increased recharge.*" |
| 15 | P.7, Figure 2: the reader can only assume that in panel (a) the bar chart indicates precipitation and the cross/lines stream flow. Please explicitly describe that in the caption. For panels (c) and (d) please avoid using red and green shades in the same figure: ~15% of your readers will be red-green colour blind. | The precipitation and streamflow are now indicated in the caption - thanks for the prompt. The image was run through the eight colour-blind variants provided by the simulator at https://pilestone.com/pages/color-blindness-simulator-1#, paying particular attention to deuteranopia (red-green). The colours adopted are distinguishable in all cases, so the figure colours are unchanged. |

| | Comment
*(line numbers refer to the original HESS-D submission)* | Response
*(line numbers refer to the updated manuscript, non-track-changes version)* |
|---|---|---|
| 16 | P.8, l.195: is this so? Then these studies may be based on a somewhat incomplete definition of "drought". In any case, please provide supporting reference for this statement or remove it. | This paragraph is now deleted as suggested (line 203). The key point—that all seasons matter and particularly the seasonal wet periods—is implied by the preceding text. |
| 17 | P.8, l.200-202: this is a repetition of P.5, L.135-139 and Figure 1. Can be omitted. | Deleted as suggested (Line 204). |
| 18 | P.8, l.218ff: although the explanation of the "two water world" (TWW) hypothesis is correct here, it does not at all support your argument here. TWW describes actual water ages and the related transit and residence time distributions that are largely controlled by the physical transport *velocities* of individual water molecules. Here, seasonal or annual water budgets are considered. These are instead controlled by *response times* which are regulated by the propagation of pressure waves at given *celerities* (see McDonnell and Beven, 2014; and Figure 2a in Hrachowitz et al., 2016). The reference to the TWW is therefore unsuitable and actually incorrect here. Instead what you describe here ("…water can move to the stream only after soil pores have been replenished") is the functioning and role of water storage following the concepts of Field Capacity and Permanent Wilting Point. Please remove any reference to TWW here. | We have rewritten this section with no reference to the Two Water Worlds hypothesis, as suggested, and changed the text to mention field capacity as appropriate. The revised text (Line 221) is:

*"In this study area, autumn is the time when many catchments "wet up" ahead of the main flow generating period in winter (June-August) and early spring. In terms of hydrological processes, tightly bound soil water, once seasonally depleted, likely gets priority in the seasonal re-wetting process (eg. Brooks et al., 2009), since mobility of water moving to the stream is restricted until after the soil pores have been replenished and the soil reaches field capacity. During the Millennium Drought, lower autumn rainfalls may have extended the soil-pore-filling period later into the year, diminishing the period of flow generation, a concept supported by Wasko et al. (2020)."* |
| 19 | P.11, l.240 (also Table 1): not sure how HPE02 is different to HPE02. Both are effectively the result of higher water deficits (i.e. lower water content below Field capacity) due to more pronounced dry periods. In both cases, the water deficit is not overcome – water content in soils remains below field capacity and is therefore held against gravity instead of being released directly (or via groundwater) to the stream. Please clarify the difference between HPE01 and HPE02. In addition, please specify more precisely what is meant by "initial losses". Do you mean water stored in soils and eventually released as transpiration or (to a minor degree) soil evaporation? Where else could this water be lost to? | Agreed that there is considerable overlap in hydrological processes, but the meteorological drivers and timescales are distinct, meriting different HPEs. As requested, we have attempted to better distinguish the two with the following text. The "initial losses" phrase has been removed (a common term in Australia but perhaps not elsewhere!). Line 237 now reads:

*"Although HPE01 and HPE02 are similar, HPE01 is focussed on the seasonal dynamic of refilling the soil moisture deficit accrued over the entire dry season, whereas HPE02 is concerned with shorter timescales and occurred in all seasons, albeit to different degrees (Figure 3b). With longer periods between significant rainfall events, soil moisture deficits likely arose from continued evaporation and/or transpiration, which then needed to be filled prior to the resumption of streamflow (similar to HPE01). Because of this, a greater proportion of event rainfall would be used to fill this deficit rather than converted to streamflow."* |

| | Comment
*(line numbers refer to the original HESS-D submission)* | Response
*(line numbers refer to the updated manuscript, non-track-changes version)* |
|---|---|---|
| 20 | P.12, l.270: "activation of long-term…processes" is very vague. Please try to be more specific. How does the reader have to imagine that? Is it really a binary phenomenon (active vs. deactivated?) or is it a process that gradually and proportionally becomes more relevant and visible the drier the system becomes? | Agree that this statement was too binary; it is now worded as "*for which long-term or long-memory process responses were less apparent*" (line 279). Agree that the statement on its own is very vague. To make this less vague, we have now inserted direct cross references to subsequent HPEs which describe such processes (line 279):

"*This broad, overarching HPE is incomplete without further detail as to which processes are in view, but this is held off to later HPEs (specifically HPEs 9, 15, 16, 17 and 18).*" |
| 21 | P.12, l.279: please avoid absolute terms such as "veracity". The best we can do in large-scale hydrology is to test and evaluate our hypothesis. | Agreed. This term has now been deleted (Line 284) and we double checked to ensure there were no remaining cases of "veracity" and similar term "validity" |
| 22 | P.12, l.293ff: perhaps good to refer to Jaramillo et al. (2018), who provide a good illustration of the counteracting effects of fertilization. | Added, thanks. (Line 300) |
| 23 | P.13, l.304: it is not quite clear to me (1) why HPE07 falls under "vegetation conditions" and not under "meteorological dynamics", (2) what the difference between HPE07 and HPE04 is, as both radiation and temperature are major controls on evaporative demand (HPE04, i.e. potential evaporation) and (3) why in the one sentence description the focus of HPE07 is radiation and temperature, while in the text just above, the turbulence differences are the actual core of this HPE. Please clarify. | We feel it is worth distinguishing between HPE04 and HPE07 because HPE07 provides a mechanism whereby the atmospheric drivers of PET can have a larger impact in grasslands compared to forested catchments, simply by virtue of the characteristics (shape) of the vegetation itself. This is important because shifted catchments were dominated by grasslands. As explained later in the paper (Section 4.1) the match in spatial distributions is an important contributor to plausibility. We have added the following text (Line 309):

"*[This HPE] is listed under vegetation because it is the characteristics of the vegetation itself which is key to the process and leads to spatial differences in this effect.*" |
| 24 | P.13, l.322: agreed, but they will not only intercept water on their foliage and thus allow evaporation, but they will also continue to transpire water. Please try to be more specific here. | Apologies for poor use of the word "intercept". This has been changed and the paragraph now refers to transpiration and interception as separate processes (Line 345). |
| 25 | P.13, l.324-324: perhaps explicitly refer to C4 grass here. | C4 has been added to the wording of the HPE (Line 331). |
| 26 | P.14, l.331: why only in south-east Australia? I would suspect this to be a general phenomenon. | True. The words "in south east Australia" have been deleted, broadening the spatial scope of the statement (Line 347) |
| 27 | P.14, l.336: not sure what is meant by "assuming some other mechanism for lower stream flow,…". Please clarify and rephrase. | We have inserted the following paragraph before the one in question (Line 339):
*"The next HPE is one of three (namely HPEs 10,* |

| | Comment
*(line numbers refer to the original HESS-D submission)* | Response
*(line numbers refer to the updated manuscript, non-track-changes version)* |
|---|---|---|
| | | *11 and 12) which are phrased differently. As noted earlier, forested catchments rarely shifted, whereas grasslands commonly shifted. The most straightforward way to explain this spatial pattern is to posit a hypothesis or hypotheses that act in grasslands catchments and not forested catchments. A less straightforward, but still possible explanation is to posit that the mechanism causing the shifts acted on all catchments regardless of land cover, and then a second mechanism, specific to forested catchments, counteracted the first to produce the observed spatial pattern. HPEs 10, 11 and 12 are in this latter category, and this is why the HPE wording is preceded by the words "assuming some other mechanism for lower streamflow...".* |
| 28 | P.14, l.344: see above | see above |
| 29 | P.15, l.362: see above | see above |
| 30 | P.15, l.365ff: Not clear what the actual processes involved here are meant to be. In my understanding the idea is that upwelling groundwater, persistent over several decades dissolved salts from deeper parts of the soil and moved it to near-surface layers. If this is so, I do not understand how lowering the groundwater tables would reduce salinity over a relatively short period: solute movement in the subsurface, in particular during dry conditions (i.e. droughts!) is characterized by very slow transport velocities and long-term legacy effects (e.g. Basu et al., 2010; Hrachowitz et al., 2015). Instead, solutes moved into near-surface layers during wet conditions will frequently undergo evapo-concentration effects (e.g. Hrachowitz et al., 2015), thus temporarily making conditions for plant growth even more unfavourable. | Apologies, this was poorly/incorrectly worded, and has now been extensively reworded. The thrust of the HPE remains similar (ie. salinity-affected and waterlogged vegetation in poor health pre-drought; healthier (and additional) vegetation during drought) but a crucial detail added is that the type of plants used in saline/waterlogged areas were more tolerant to these conditions and thus capable of flourishing where the previous vegetation had not. While we agree soil salinity may persist for many years, the ill-effects of the waterlogging itself on vegetation are significant and should also be acknowledged. The revised wording is (line 380):

*"This HPE spans vegetation and groundwater processes. Prior to the Millennium Drought, multi-decadal wet conditions combined with historic deforestation led to high water tables and waterlogging in many areas, with naturally-occurring soil salt brought to the surface (eg. Cartwright et al., 2004). The waterlogged and saline topsoil reduced the health, and thus water use, of vegetation (Lambers, 2003). Waterlogged and salt-affected areas were planted with tolerant grasses or shrubs, while trees were added to the landscape in groundwater recharge areas or upslope of discharge areas (Schofield, 1992; Marcar, 2015). These revegetation efforts were focussed on the late 1980s and 1990s, and thus the timing of maturation of much of this vegetation would have coincided with the onset* |

| | Comment *(line numbers refer to the original HESS-D submission)* | Response *(line numbers refer to the updated manuscript, non-track-changes version)* |
|---|---|---|
| | | *of the Millennium Drought (commencing late 1990s). Further, the drought onset lowered water tables, removing waterlogging as a stressor of vegetation (although it is noted that legacy salinity may persist for many years and even intensify during drought due to evapo-concentration; see eg. Hrachowitz et al., 2015). HPE13: Streamflow was lower than expected because vegetation recovered from prior waterlogging, and because of the maturation of vegetation planted earlier to combat salinity."* |
| 31 | P.15, l.376: the use of the term "interception" here and elsewhere in the manuscript is ambiguous. Please note that "interception" has a very specific meaning in hydrological literature (e.g. Miralles et al., 2020; Savenije, 2004). What is specifically meant here? A process that retains water on the canopy, foliage or near-surface soil layers to supply water for the "evaporation" process or is it used in a more general way to also include "interception" in in the root zone to supply root-water uptake for transpiration? If it is the latter, I strongly recommend to rephrase to avoid misunderstandings. | Apologies, the word "interception" is used in a more general way in Australia, as per your comment. We agree that this is unhelpful here since it has only one definition in the hydrological processes literature. In response, every instance of the word "intercept" in the manuscript has now been replaced by an alternative word or phrase, except in the one case (HPE14, line 393) where it is consistent with the specific meaning stated by the reviewer. |
| 32 | P.15, 377ff: HPE15 is described only in very broad and vague terms. I agree, that the actual processes here may be unknown. However, in such a case, I am not sure if a meaningful hypothesis can be formulated, because a meaningful hypothesis always needs to be testable, otherwise it cannot be qualified as hypothesis. Please try to more explicitly specify this hypothesis or remove it, as it may be indistinguishable from most of the other vegetation-related HPEs here. | The aspect that distinguishes this HPE from all the others is the hypothesis that catchments have multiple stable states of behaviour. As per our added text (line 409): "*It should be noted ... that none of the preceding HPEs inherently give rise to multiple stable states, and thus such a hypothesis must be stated separately, which is why it is given its own HPE here.*" Although the exact processes are unknown, it is important to mention this hypothesis because it is put forward by one of the two main studies of the hydrological shifts within the study area (namely Peterson et al., 2021). Although we considered removing it as suggested, we note that Reviewer #1 emphasised that this should be highlighted as a "potential perspective to explain the change in behaviour". With both a reviewer and a prominent prior study affirming this hypothesis, we felt it was best to retain it.

We responded to this comment by making it clearer in the text that the distinguishing feature is the multiple stable states and by explaining this in greater detail. Almost all of the following text is new to this section (Line 397): |

| | Comment
*(line numbers refer to the original HESS-D submission)* | Response
*(line numbers refer to the updated manuscript, non-track-changes version)* |
|---|---|---|
| | | *"Peterson et al. (2021) suggest vegetation behaviour may explain the apparent appearance of hydrological shifts. Further, they suggest that their findings (as summarised in Section 2) are "consistent with … watersheds having multiple stable states [of behaviour] and a finite resilience". In this view, the hydrological shift towards a lower rainfall-runoff relationship corresponds to a transition from one stable state to another (see Peterson et al., 2009). It is noted that the word "state" here has a different meaning to its common hydrological usage (eg. "state variable"), as it refers to the behaviour of a system being organised into discrete preferential states (eg. the "wet" and "dry" states of D'Odorico and Porporato (2004)), with intermediate conditions having low probability of occurrence. The Millennium Drought was an extreme disturbance that might have pushed several catchments from a wetter to a drier state, with the drier state otherwise seldom being apparent in a catchment's behaviour."* |
| 33 | P.16, l.392: not sure if "activated" is the most suitable term here (see one of the comments above) | This now reads: "*the long duration of the Millennium Drought increased the role of long-term (long-memory) processes in determining hydrological behaviour*" (Line 418) |
| 34 | P.16, l.403: what is "diffuse discharge"? | Our meaning was that diffuse discharge is loss of groundwater via soil evaporation over an extended area. On reflection, we have removed the reference to diffuse discharge because it is not relevant to the paragraph's focus on interactions with streamflow. |
| 35 | P.16, l.411: please specify "discharge areas" | This is now changed to "partial saturated areas" in line with Dunne and Black (among others) (Line 429). |
| 36 | P.16, l.414: "interception by transpiration"? Please see comment above. | Reworded to (Line 439):

"*potentially increasing the probability of being diverted to transpiration (Jensen et al., 2017; 2018) rather than contributing to streamflow.*" |
| 37 | P.18, l.435: Hulsman et al. (2021a) similarly found supporting evidence for the importance of upland groundwater sustaining alluvial evaporation/transpiration in a large scale study in the Zambezi basin. Perhaps nice to include as reference. | Thanks for the relevant reference, it has been included as suggested (Line 461) |

| | Comment
*(line numbers refer to the original HESS-D submission)* | Response
*(line numbers refer to the updated manuscript, non-track-changes version)* |
|---|---|---|
| 38 | P.18, l.444-446: explicitly mention the role of evaporation/transpiration here to be more specific. | This now reads (Line 471): *"HPE16: Streamflow was lower than expected because water slowly drained from hillslopes (contributing to water table decline and reduced GW-SW interaction) to nearby alluvial areas, and was subsequently lost via evapotranspiration."* |
| 39 | P.18, l.449: potentially very relevant and ofter overlooked process. Bouaziz et al. (2018), Condon et al. (2020), Hulsman et al. (2021b) but also the authors themselves (Fowler et al., 2020) provide different recent perspectives on the potential importance of this process. Frisbee et al. (2012) also provide an excellent synthesis and illustration (Figure 3 therein!). Would be good to add at least some of these references here to provide a stronger context and background for the reader. | You are right that the references are too sparse here. All of these have been added, and the section is much improved, thank you (Line 474). |
| 40 | P.19, l.461: "[…] some systems […] that are more extensive […] have longer response times […]". Really? I am not sure this reasoning is generally valid. In larger systems the average flow distance in the subsurface domain, the controlling factor on response times at time scales > one month, is in most environments very similar to those of smaller, headwater systems. In other words, no matter if you are anywhere upstream or downstream in the landscape, the distance to the next river will not be that different, due to the fractal, scale-invariant nature of river networks (e.g. Rodriguez-Iturbe and Rinaldo, 2001). Please provide a reference or remove. | This now reads (Line 490)

"*In south-east Australia, groundwater systems which cover larger areas tend to have longer response times (Walker et al., 2003).*"

We have added the place-based qualification and the reference (ie. the underlined words).

We do not include the following detail in the text, but Figure 2 of this reference, along with accompanying text, suggests local flows systems respond after "a few years", intermediate after 50-100 years and regional systems after 100+ years. This information is specific to Australia, so this scaling with size possibly doesn't apply in groundwater systems elsewhere. |
| 41 | P.19, l.485ff: this HPE needs more explanation. It is not clear why the vadose zone should indeed be drier. The reasoning here, as far as I understand, is that declining GW tables result in deeper vadose zones. Ok. In these parts of the subsurface, all the water that cannot be held against gravity (water above Field Capacity) will released and "follow" the falling GW table. The remainder, i.e. soil water content at Field Capacity, will largely be held against gravity. This water can only be released by evaporation or plant water uptake for transpiration. Assuming that in many locations the GW-table is below the root zone, plant water uptake drops out as potential process to remove water. However, soil water at depth below 20- 30cm can also not be evaporated at very high rates (e.g. Brutsaert, | Apologies, the original HPE was not well worded to capture the underlying idea. The key clarification we would make with respect to the reviewer's comment is that the drying here refers to moisture contents greater than field capacity but less than saturation. Thus, we agree that the soil moisture is unlikely to fall below the level which is held against gravity in the absence of the plant roots.

We have completely re-worded this HPE. It now reads (Line 515):

*"Regardless of the cause of the falling water table, the result was a thicker vadose (unsaturated) zone as the groundwater subsided, leaving behind newly unsaturated layers. This* |

| | Comment
*(line numbers refer to the original HESS-D submission)* | Response
*(line numbers refer to the updated manuscript, non-track-changes version)* |
|---|---|---|
| | 2014). The deeper, the less relevant soil evaporation will be due to the limited diffusive gas/vapour exchange with the surface (there is no wind in the soil pores for turbulent exchange!). Soil deeper below the root zone will thus frequently be close to Field Capacity, as the water cannot be released with gravity only and very limited evaporation. It would be great if you could provide a more detailed description of your hypothesis that soils in a deeper vadose zone can be drier (i.e. below Field Capacity – because if they remain at or above field capacity, they will be hydraulically and hydrologically irrelevant. In that case, all the water that enters this zone from above will be again released as it cannot be held, i.e. dS/dt ~ 0 over time scales larger than a few days). | *HPE is concerned with the rate at which these layers drain from saturation to field capacity. Well-drained near-surface soils are generally considered to drain quickly, on the order of days (eg. Cassel and Nielsen, 1986). However, drainage may be slower for deeper layers (particularly those composed primarily of bedrock), possibly associated with spatially discontinuous preferential flow mechanisms including funnelled flow (eg. Nimmo, 2020) and unstable flow (eg. Jury et al., 2003). These deeper unsaturated layers might temporarily absorb some recharge, delaying its passage downwards to saturated zones, and spreading the recharge signal over time. Whereas a rapid recharge event might trigger rapid groundwater discharge to the stream, the dampening of the recharge signal might increase the probability of water being diverted to transpiration (particularly by vegetation close to water courses and drainage lines) that would have otherwise contributed to streamflow, thus increasing transpiration per unit precipitation. HPE19: Streamflow was lower than expected due to delaying of recharge by the enlarged unsaturated zone, thus increasing opportunities for transpiration."* |
| 42 | P.19, l.487-488: I suspect you mean that the "infiltration capacity" declines with increasing wetness as described by Darcy-Richards. In contrast, "hydraulic conductivity" typically increases with increasing wetness! | Yes, apologies for the error, but this HPE has now been completely reworded (see above) so the issue is moot. |
| 43 | P.20, l.495: that cracked soils result in less runoff is of course not impossible. However, also the opposite, the importance of cracks as preferential flow pathways, is frequently observed and documented (e.g. Zehe at al., 2013). Please adjust the hypothesis accordingly. | True. We have added the words (Line 536)

*"likewise, soil cracking may provide an avenue for preferential flow, thus increasing runoff (eg. Beven and Germann, 1982)."* |
| 44 | P.20, l.497-498: "Streamflow was lower […] because […] higher infiltration […]". This does not quite add up for me. Water that infiltrates surely does not disappear. Was the assumption that most of it will be held in the soil and evaporated/transpired instead of recharging the GW? | We have added an extra qualification to this explanation for this HPE. The altered sentence with the new qualification underlined, is (Line 530):

*"For example, soil cracking may lead to increased infiltration, more evaporation and less surface runoff (eg. Arnold et al., 2005), provided the infiltrated water remains in the soil rather than recharging groundwater."* |
| 45 | P.21, l.529: perhaps better to replace "interception" by something like "retention and subsequent evaporation/transpiration" | "Interception" has been replaced by "harvesting of water" (Line 570) since this is a more appropriate term for farm dams. We also add to |

| | Comment *(line numbers refer to the original HESS-D submission)* | Response *(line numbers refer to the updated manuscript, non-track-changes version)* |
|---|---|---|
| | | text (Line 566) to clarify that "*Flow impacts [from farm dams] occur due to both extracted water (for stock, domestic or irrigation use) and evaporated water being unavailable to flow downstream.*" |
| 46 | P.23, l.584: should this read as "105 "? | Thanks for noticing this error, it has been corrected to $10^5$ (Line 627) |
| 47 | P.23, l.600: please also explicitly mention the three plausible hypotheses here and not only in the table. | Done. We also added a list of the 11 HPEs in the category in-between, along with the reasons why they did not qualify for the top category (Line 648). |
| 48 | P.24, HPE08: "[…] it is doubtful whether modest historical CO2 increases could have caused larger changes […]". Without any further data support, more detailed reasoning and/or references this remains largely speculation and cannot be used as hypothesis test. | This is a fair criticism; we have added the following text (Row 8 of Table 2): "*Relative AET changes of the order of 5-10% would be needed, and changes of such magnitude have indeed been reported in the literature (eg. Figure 4 of Morgan et al., 2004), but only in response to a high level of $CO_2$ enrichment (600 $\mu l\ l^{-1}$), which corresponds to a much higher concentration of $CO_2$ than those seen historically.*" |
| 49 | P.25, HPE13: in the light of my comments above, the reasoning here (low water table) is not very convincing. | See above row 30 in this table - with the clarification that waterlogging is in focus, not just salinity; and the further focus on salt-tolerant species, we feel the text of Table 2 is reasonable. |
| 50 | P.26, HPE19: see above. A deeper vadose zone will only allow further water retention if water from pores is being extracted by soil evaporation and or transpiration. Otherwise the zone will, on average, remain close to Field Capacity and act as a hydraulically and hydrologically passive part of the soil. In other words, it will cause some delay in the water percolating through this zone, but it will not provide additional "storage", i.e. on time scales of more than a few days dS/dt~0. | See row 41 of this table. |
| 51 | P.33, 684ff: see above. Also, the term "interception" is not suitable here | In line with row 41 of this table, this text (Line 752) now has an altered focus that doesn't use the word "interception". |